# Engineering a synthetic energy-efficient formaldehyde assimilation cycle in *Escherichia coli*

Tong Wu [1,2], Paul A. Gómez-Coronado [1,3], Armin Kubis[1], Steffen N. Lindner[1,2], Philippe Marlière[4], Tobias J. Erb [3], Arren Bar-Even [1] & Hai He [1,3] ✉

One-carbon (C1) substrates, such as methanol or formate, are attractive feedstocks for circular bioeconomy. These substrates are typically converted into formaldehyde, serving as the entry point into metabolism. Here, we design an erythrulose monophosphate (EuMP) cycle for formaldehyde assimilation, leveraging a promiscuous dihydroxyacetone phosphate dependent aldolase as key enzyme. In silico modeling reveals that the cycle is highly energy-efficient, holding the potential for high bioproduct yields. Dissecting the EuMP into four modules, we use a stepwise strategy to demonstrate in vivo feasibility of the modules in *E. coli* sensor strains with sarcosine as formaldehyde source. From adaptive laboratory evolution for module integration, we identify key mutations enabling the accommodation of the EuMP reactions with endogenous metabolism. Overall, our study demonstrates the proof-of-concept for a highly efficient, new-to-nature formaldehyde assimilation pathway, opening a way for the development of a methylotrophic platform for a C1-fueled bioeconomy in the future.

The biotechnological production of fuels and chemicals currently relies on sugars and biomass-derived carbohydrates as raw materials. However, this practice faces various challenges, including competition with human food production, increasing land use, threatening biodiversity, as well as laborious biomass processing, which limits scalability of this approach[1,2]. In contrast, one-carbon (C1) compounds, such as methanol or formate, are favorable alternatives, because they do not compete with food production, are widely available, relatively cheap, and can be produced in a sustainable fashion directly from $CO_2$[3,4]. This makes C1 substrates promising feedstocks for a green and circular bioeconomy and biotechnology[5–8].

In the assimilation of C1 substrates, formaldehyde plays a central role as bridging metabolite, because it is highly reactive and ready to participate in various biochemical reactions with a diverse set of molecules, such as thiols, amines, pterins, and especially (phospho-)

sugars[9–11]. Nature has developed different metabolic pathways for the assimilation of C1 substrates, which can be generally divided in two parts. First, the C1 substrate (i.e., methanol or formate) is converted into formaldehyde, which is then assimilated through downstream reactions. These pathways include the ribulose monophosphate (RuMP) cycle, the xylulose monophosphate (XuMP) cycle, or the serine cycle, which typically occur in specialized microorganisms, so-called methylotrophs. While there has been some progress in metabolic engineering of natural methylotrophs[12–14], the lack of genetic tools is still a major constraint in accelerating metabolic engineering efforts in these organisms to broaden their product spectrum and/or improve their bioproduction capabilities.

The lack of genetic accessibility of methylotrophs has motivated researchers to engineer C1-assimilation pathways in established microbial platform organisms, such as *Escherichia coli*[6,8,15]. One example

[1]Max Planck Institute of Molecular Plant Physiology, Am Mühlenberg 1, 14476 Potsdam-Golm, Germany. [2]Charité—Universitätsmedizin Berlin, corporate member of Freie Universität Berlin and Humboldt Universität zu Berlin, Institute of Biochemistry, Charitéplatz 1, 10117 Berlin, Germany. [3]Max Planck Institute for Terrestrial Microbiology, Karl-von-Frisch-Str. 10, 35043 Marburg, Germany. [4]TESSSI, The European Syndicate of Synthetic Scientists and Industrialists, 81 rue Réaumur, 75002 Paris, France. ✉e-mail: hai.he@mpi-marburg.mpg.de

is the RuMP pathway which has been extensively studied and eventually implemented into *E. coli* to allow synthetic methylotrophy of this organism[16–21]. Other efforts focused on transplanting engineered (natural) pathways, such as a modified serine cycle[22], the homoserine cycle[23], or the reductive glycine pathway[24] in *E. coli*. Moreover, synthetic pathways based on new-to-nature enzymatic activities, such as formolase[25,26] or 2-hydroxyacyl-CoA lyase[27,28], were also designed and tested. Finally, several additional pathways have been identified and proposed through computational efforts[29].

In this study, we present a synthetic formaldehyde assimilation pathway, the erythrulose monophosphate (EuMP) cycle. The EuMP cycle is devised following a mix-and-match approach[30], relies on a promiscuous aldolase reaction with formaldehyde, and is expected to show the same energetic efficiency as the RuMP cycle. We demonstrate activity of the core reactions of the EuMP cycle in *E. coli* using a modular and selection-based engineering strategy[31]. Applying adaptive laboratory evolution (ALE) allows us to integrate the core modules of the EuMP cycle with endogenously existing reactions of the pentose phosphate pathway and glycolysis. Whole genome sequencing and reverse-engineering reveal the molecular basis for implementation of the EuMP modules in *E. coli*. Overall, our study demonstrates the feasibility of a new-to-nature formaldehyde assimilation pathway for a future C1-based biomanufacturing.

## Results

### Design of the erythrulose monophosphate cycle

The EuMP cycle (Fig. 1a) is centered around a dihydroxyacetone phosphate (DHAP) dependent formaldehyde fixation reaction and connects tetritol catabolism[32,33] with the pentose phosphate pathway. The key reaction, catalyzed by erythrulose 1-phosphate synthase (EPS, or erythrulose 1-phosphate formaldehyde-lyase), condenses formaldehyde with DHAP yielding erythrulose 1-phosphate (Eu1P). Depending on the enzyme stereoselectivity, the product could be L-Eu1P or D-Eu1P, the latter one can be converted by D-erythrulose 1-phosphate 3-epimerase (EryC) into L-Eu1P. Following by sequential isomerization reactions catalyzed by L-erythrulose 1-phosphate isomerase (LerI) and D-erythrulose 4-phosphate isomerase (DerI), L-Eu1P is transformed to D-erythrose 4-phosphate (E4P), which enters the pentose phosphate pathway. With reactions of the pentose phosphate pathway and glycolysis, DHAP can be regenerated, thus closing a cyclic pathway. Overall three turns of the cycle provide one molecule of glyceraldehyde 3-phosphate (GAP) from three molecules formaldehyde.

The EuMP cycle can operate in four different architectures. Similar to the natural RuMP pathway[4], the four different architectures are based on four different possibilities of converting E4P to DHAP and GAP (Supplementary Fig. 1). The conversion of E4P to DHAP and GAP can be generally separated in two stages, i.e. a carbon rearrangement stage and a C6-sugar cleavage stage. The carbon rearrangement stage that converts E4P to fructose 6-phosphate (F6P) comes in two variants: a transaldolase (TAL) or sedoheptulose 1,7-bisphosphatase (SBP) route; in the C6-sugar cleavage stage, F6P is subsequently split into GAP and DHAP, which can also proceed through two different routes, either through fructose 1,6-bisphosphate aldolase (FBA) or the Entner-Doudoroff (ED) route. The four different combinations result in different energetic costs, as indicated in Supplementary Fig. 1. Our study focuses on the TAL/FBA variant of EuMP cycle (Fig. 1a and Supplementary Fig. 1a), which is the most energy efficient design. The in vivo thermodynamic feasibility of this design was confirmed with computational modeling by Max-min Driving Force (MDF)[34] (Supplementary Fig. 2).

Compared to the three naturally existing formaldehyde assimilation pathways, the EuMP cycle has the same ATP cost as the RuMP cycle, which is the most efficient natural C1 strategy[4]. For one molecule of GAP, the EuMP cycle and the RuMP cycle consume only one ATP

molecule and no reducing power. This is substantially less compared to the XuMP pathway and the serine cycle, which require three molecules of ATP per GAP or acetyl-CoA, respectively. The serine cycle needs additional NAD(P)H when producing GAP (Fig. 1b). We further used flux balance analysis to compare the maximal yields of biomass and the 12 building block precursors in bioproduction[35] (see "Methods"). As expected, the EuMP cycle has the same yields as the RuMP cycle in biomass as well as precursors, supporting the highest yields in 11 out of 12 precursors (Fig. 1c).

To demonstrate the activity of the EuMP cycle in vivo, we decided to follow a step-wise, modular engineering approach using the model organism *E. coli* as host[31]. To that end, we split the EuMP cycle into four modules (M1–M4) (Fig. 1a): the formaldehyde assimilation module (M1) that converts formaldehyde and DHAP to L-Eu1P; the erythrulose phosphate isomerization module (M2) that transforms L-Eu1P to E4P; the carbon rearrangement module (M3) in which the non-oxidative pentose phosphate pathway reactions reconfigure E4P to GAP and F6P; and the C6-sugar cleavage module (M4) that cleaves F6P for DHAP regeneration and GAP as product. As the reactions of the latter two modules are part of canonical central carbon metabolism and present in *E. coli*, the formaldehyde assimilation and erythrulose phosphate isomerization modules (M1 and M2) are key to successfully establish synthetic EuMP in vivo. Hence, we aimed to reconstruct these modules in dedicated selection strains for demonstrating their activities in vivo.

### Validation of the erythrulose phosphate isomerization module (M2) in *E. coli*

We first aimed at realizing the erythrulose phosphate isomerization module (M2) by employing LerI and DerI that together transform L-Eu1P to E4P. We used a ΔtktAB strain (Fig. 2a and Supplementary Table 1), which is blocked in E4P biosynthesis through the knockout of both transketolase (TKT) genes, *tktA* and *tktB*[36]. To grow on minimal medium, this strain requires E4P supplements (E4P_suppl.: 1 mM shikimic acid, 1 μM pyridoxine, 250 μM tyrosine, 500 μM phenylalanine, 200 μM tryptophan, 6 μM 4-aminobenzoic acid, 6 μM 4-hydroxybenzoic acid and 50 μM 2,3-dihydroxybenzoic acid)[18], because E4P is a precursor of several essential cellular building blocks and cofactors, such as shikimate derived aromatic amino acids and vitamins as well as pyridoxal phosphate. We overexpressed module M2 (i.e., LerI and DerI), together with other tetritol catabolism enzymes, erythritol/L-threitol dehydrogenase (EltD) and L-erythrulose 1-kinase (LerK) from plasmid pKIID in the ΔtktAB strain (Supplementary Table 1). EltD and LerK can convert L-threitol and erythritol, two tetritols, into L-Eu1P and D-erythrulose 4-phosphate, respectively[33]. If active, M2 further transforms these products into E4P, which would allow for growth from the two tetritols in the absence of E4P_suppl. in the selection strain (Fig. 2a).

Indeed, while both tetritols were not able to directly replace E4P_suppl. to feed the ΔtktAB strain (dashed lines in Fig. 2b), plasmid pKIID enabled growth of the strain on 10 mM L-threitol and erythritol, which were also essential for growth (solid green and pink lines in Fig. 2b). This suggests that L-threitol and erythritol can be converted into E4P with the enzymes from pKIID, indicating that the tetritol catabolism route and particularly the module M2 via LerI and DerI was functional in *E. coli*. (We note that the lag phase seems to result from tetritol catabolism intermediates, and it appeared only when both the pKIID plasmid and L-threitol or erythritol were presented (Supplementary Fig. 3)).

### Establishing erythrulose 1-phosphate synthase (M1) in vivo extends M2

We next aimed at demonstrating the core reaction of EuMP, erythrulose 1-phosphate synthase (EPS) in vivo. This reaction, EC 4.1.2.2, was reported in the 1950s to produce L-erythrulose 1-phosphate (L-Eu1P) by extracts from rat liver[37] and Swiss chard leaves[38]. Later on, this

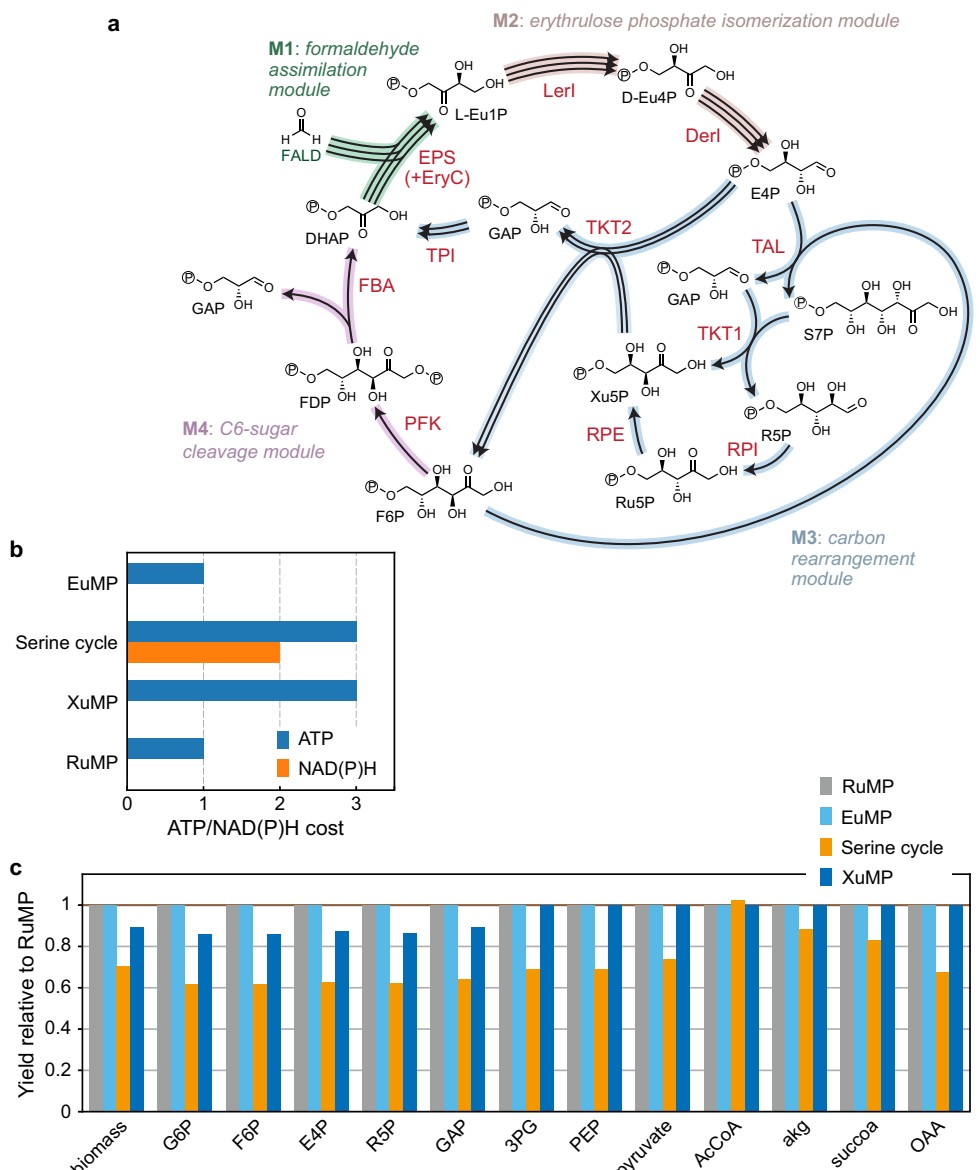

**Fig. 1 | The synthetic erythrulose monophosphate (EuMP) cycle. a** A schematic representation of the EuMP cycle and its modularization. Pathway enzyme names are shown in red. The four modules for metabolic engineering are highlighted in different color codes. Overall, the pathway transforms three molecules formaldehyde (FALD) into a molecule of glyceraldehyde 3-phosphate (GAP). **b** ATP and reducing equivalent costs of the EuMP cycle and the three natural formaldehyde assimilation pathways, RuMP (ribulose monophosphate pathway), XuMP (xylulose monophosphate pathway) and serine cycle. The costs are for production of acetyl-CoA in terms of serine cycle and for GAP in terms of other pathways. **c** The EuMP pathway is equally efficient as the natural RuMP for most precursor production. See Methods for the flux balance analysis. The values were normalized to the yields via the RuMP cycle. 3PG 3-phosphoglycerate, AcCoA

acetyl-coenzyme A, akg 2-ketoglutarate, DHAP dihydroxyacetone phosphate, DerI D-erythrulose 4-phosphate isomerase, E4P D-erythrose 4-phosphate, EPS D-erythrulose 1-phosphate synthase, EryC D-erythrulose 1-phosphate 3-epimerase, D-Eu1P D-erythrulose 1-phosphate, L-Eu1P L-erythrulose 1-phosphate, D-Eu4P D-erythrulose 4-phosphate, F6P fructose 6-phosphate, FALD formaldehyde, FBA fructose-bisphosphate aldolase, FDP fructose-1,6-bisphosphate, G6P glucose 6-phosphate, GAP glyceraldehyde 3-phosphate, LerI L-erythrulose-1-phosphate isomerase, OAA oxaloacetate, PEP phosphoenolpyruvate, PFK phosphofructokinase, R5P ribose 5-phosphate, RPE ribulose-5-phosphate 3-epimerase, RPI ribose-5-phosphate isomerase, Ru5P ribulose 5-phosphate, S7P sedoheptulose 7-phosphate, succoa succinyl-coenzyme A, TAL transaldolase, TKT1 and TKT2 transketolase, TPI triosephosphate isomerase, Xu5P xylulose 5-phosphate.

reaction was shown to be catalyzed by class I fructose 1,6-bisphosphate aldolase from rabbit muscle[39], *Staphylococcus carnosus* as well as *Staphylococcus aureus*[40,41]. In addition, other studies reported that L-rhamnulose 1-phosphate aldolase (RhaD) and L-fuculose 1-phosphate aldolase (FucA) from *E. coli* also showed EPS activity, producing D-erythrulose 1-phosphate (D-Eu1P) from DHAP and formaldehyde[42,43]. All these enzymes (i.e., fructose 1,6-bisphosphate aldolase, RhaD and FucA) are DHAP-dependent aldolases and that the reported EPS activity is not their primary function, but rather a side activity. The promiscuity of all these enzymes is in line with the fact that aldolases

are generally relatively specific to their donor substrates while being flexible in respect to different aldehyde acceptors[44].

Apart from RhaD and FucA, *E. coli* has five more DHAP-dependent aldolases, FbaA, FbaB, GatY, KbaY and YihT. Based on their sequence similarities (Supplementary Fig. 4), we generated a phylogenetic tree to show the sequence-function relationship of all these proteins with each other. As shown in Fig. 3a, the proteins cluster according to their reported functions: FucA and RhaD cluster together and have the same acceptor (*S*)-lactaldehyde; FbaA, GatY and KbaY accept D-glyceraldehyde 3-phosphate as substrate and derive from a common

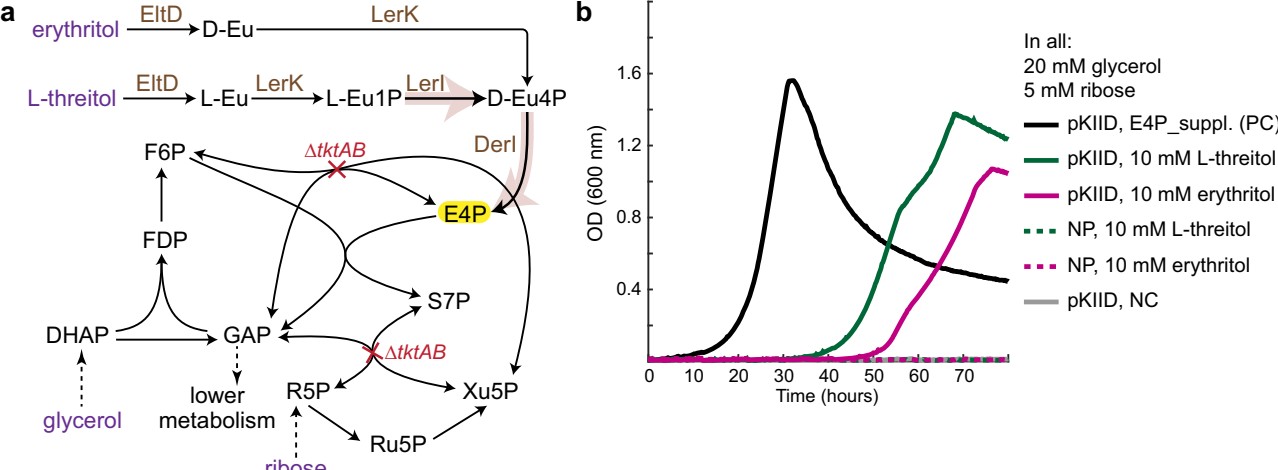

**Fig. 2 | The erythrulose phosphate isomerization module (M2) is active in *E. coli*. a** A schematic representation of the ΔtktAB selection scheme. Gene deletions are in red. Such a strain cannot synthesize the essential metabolite E4P, highlighted in yellow. Enzymes expressed from a plasmid, pKIID (Supplementary Table 1), are in brown. Carbon sources are shown in purple. D-Eu is D-erythrulose, EltD is erythritol/L-threitol dehydrogenase, FBP is fructose-1,6-bisphosphatase, L-Eu is L-erythrulose, and LerK is L-erythrulose 1-kinase. Other abbreviations are the same as Fig. 1. **b** L-threitol and erythritol can replace E4P supplements (E4P_suppl., see "Methods") only when pKIID presents. PC represents positive control, NC represents negative control of no L-threitol or erythritol, NP represents no plasmid control. Source data are provided as a Source Data file.

branch, with FbaA forming the (3 *S*, 4 *R*) product and GatY and KbaY forming the (3 *S*, 4 *S*) product; YihT and FbaB, finally, are class I aldolases that are distant from all the other enzymes, which are class II aldolases. Even though the different aldolases possess different stereochemical properties (Fig. 3a), the difference at $C_4$ becomes irrelevant when using formaldehyde as acceptor, because the product Eu1P does not contain a chiral center at $C_4$. In terms of stereochemistry at the $C_3$, RhaD and FucA are reported to produce the 3 *R* product, which would correspond to D-Eu1P when using formaldehyde as substrate, while the other five are reported to produce the 3 *S* product, which would correspond to L-Eu1P. Since GatY and KbaY are more complex enzymes and require additional, non-catalytic subunits (GatZ and KbaZ, respectively), for full activities and stability[45], we excluded those candidates for our implementation efforts. We also ruled out class I type FbaB and just kept its class II paralog FbaA, which is generally more stable[46]. Overall, we decided to use *E. coli*'s RhaD, FucA, FbaA, and YihT as candidates to realize the EPS reaction in vivo.

To test the four EPS candidates in vivo, we again used the ΔtktAB strain, which we amended by an additional knockout of the *frmRAB* operon (ΔfrmTKT, Fig. 3b and Supplementary Table 1). FrmA and FrmB are responsible for the glutathione-dependent oxidation of formaldehyde to formate. The *ΔfrmRAB* knockout abolishes this route, strongly increasing intracellular formaldehyde concentrations[18]. We overexpressed each of the four different EPS candidates together with genes *lerI, derI, eryC*, and *soxA* (Supplementary Table 1). LerI and DerI are for the module M2 that we had demonstrated earlier. EryC (D-erythrulose 1-phosphate 3-epimerase) is necessary for RhaD and FucA to convert D-Eu1P into L-Eu1P. Although EryC should not be essential when using FbaA and YihT, we still used it in these constructs, reasoning that the EryC reaction is fully reversible, $\Delta_r G' = 0$ kJ/mol[47]. Sarcosine oxidase (SoxA), finally, was used for providing formaldehyde via a reliable method from sarcosine in situ[18,48]. To focus on the demonstration of the EuMP pathway, we used this established formaldehyde production modular to set aside the inefficient production of formaldehyde from C1 substrates in this study. The overall selection scheme is shown in Fig. 3b. The ΔfrmTKT strain can grow only when M1 (i.e., the different EPS candidates and EryC in case of RhaD and FucA) is active and able to condense sarcosine-derived formaldehyde with DHAP to L-Eu1P. L-Eu1P is further processed downstream by M2 to provide E4P to the selection strain.

Strikingly, when testing the different candidates, only FucA and RhaD (overexpressed from plasmids pFEIIO and pREIIO, respectively, Fig. 3c, d), but neither FbaA nor YihT (Supplementary Fig. 5), enabled growth of the ΔfrmTKT strain on sarcosine as formaldehyde source. In these strains, growth rates and maximal ODs directly correlated with increasing sarcosine concentrations (Fig. 3c, d) and growth could not be restored without EPS (i.e., *fucA* or *rhaD*) overexpression (Fig. 3e). Moreover, omitting EryC resulted in no growth (Fig. 3f), supporting the hypothesis that FucA and RhaD yield D-Eu1P and require EryC-catalyzed D-Eu1P - L-Eu1P isomerization for formaldehyde assimilation. We attributed the small growth under sarcosine absent condition (red lines in Fig. 3c and d) to a non-depletable internal source of formaldehyde[9,11], because such growth was FucA/RhaD and EryC dependent (Fig. 3e, f and Supplementary Fig. 5). In contrast to FucA overexpression, overexpression of RhaD did not allow growth at elevated sarcosine concentrations (i.e., > 5 mM), indicating that FucA removes/assimilate formaldehyde more efficiently in vivo, concerning the toxicity of formaldehyde.

To further confirm that E4P was produced through condensation of formaldehyde with DHAP via EPS, i.e., module M1, and module M2 of the EuMP cycle, we performed [13]C-labeling trace experiments (Fig. 3g, h and Supplementary Fig. 6a). When cultivating the FucA and RhaD overexpression strains with unlabeled glycerol and *N*-methyl-[13]C labeled sarcosine, we observed single-labeled tyrosine and phenylalanine. (The small fractions of double labeling were results of natural abundance of [13]C (Supplementary Fig. 6a) and the relatively high unlabeled fractions, especially for FucA, were likely from internal source formaldehyde originated from unlabeled glycerol.) The labeling pattern of tyrosine and phenylalanine is in line with the condensation of non-labeled phosphoenolpyruvate (PEP) with single-labeled E4P (derived from DHAP and [13]C-formaldehyde) during aromatic amino acids biosynthesis. This hypothesis was further confirmed by analysis of the labeling pattern of alanine, which is directly derived from PEP and was indeed unlabeled in these strains. We note that TAL transfers carbons between F6P and GAP, resulting in unlabeled GAP and subsequently PEP. Yet the flux of TAL is net 0 (calculated by flux balance analysis, Supplementary Fig. 7a). Overall, these experiments confirmed that the EuMP core modules M1 and M2 were successfully realized in vivo.

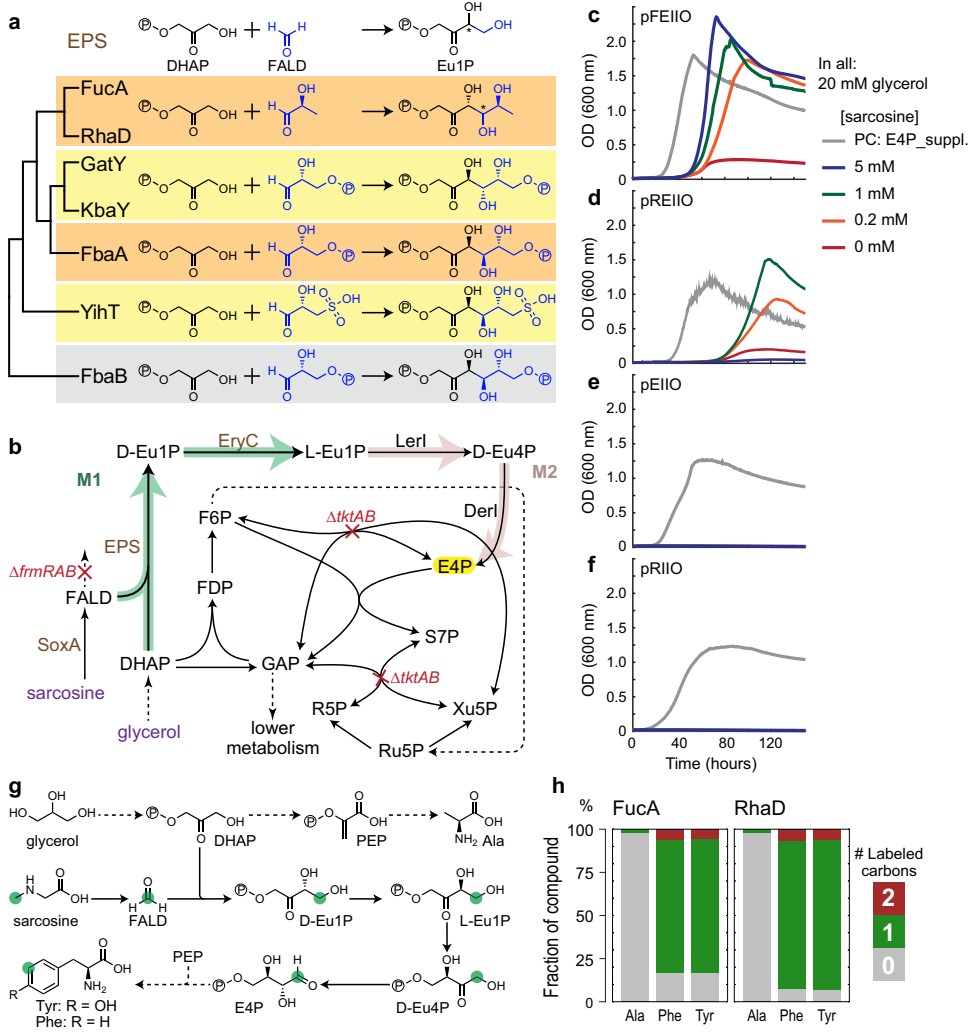

**Fig. 3 | In vivo selection of EPS in the formaldehyde assimilation module (M1).**
**a** EPS candidates from *E. coli* that are known to be DHAP dependent aldolases. Phylogenetic relationship of the proteins is based on their sequence similarity (Supplementary Fig. 4 and "Methods"). FbaB is used as an out group. **b** A schematic representation of the ΔfrmTKT selection. Gene deletions are in red. Such a strain cannot synthesize the essential metabolite E4P, highlighted in yellow. Enzymes expressed from plasmid are in brown. Carbon sources are shown in purple. SoxA, sarcosine oxidase. Other abbreviations are the same as Fig. 2. **c** FucA and (**d**) RhaD, once overexpressed together with other EuMP enzymes, restored growth of the

ΔfrmTKT strain in a sarcosine dependent manner. Strains omitted EPS (**e**) or EryC (**f**) overexpression failed to grow. Expressed plasmids (detailed in Supplementary Table 1) are shown at the top left corner of the curves. FbaA and YihT failed to support the growth, the results are shown in Supplementary Fig. 5. **g** Trace of ¹³C labeling from formaldehyde. **h** Labeling pattern of proteinogenic alanine (Ala), phenylalanine (Phe), and tyrosine (Tyr), within the strains ΔfrmTKT overexpressed FucA (pFEIIO) and RhaD (pREIIO), upon feeding with unlabeled glycerol and sarcosine-(methyl-¹³C). The accompanied results with unlabeled sarcosine are shown in Supplementary Fig. 6a. Source data are provided as a Source Data file.

## Adaptive laboratory evolution integrates carbon rearrangement module M3

We finally sought to further integrate the EuMP cycle by connecting the two active core modules (M1 and M2) with the naturally existing carbon rearrangement module (M3) in *E. coli*. To that end, we constructed a ΔFBP/GlpX strain in which the fructose 1,6-bisphosphatases of gluconeogenesis were knocked out, Δfbp ΔglpX (Fig. 4a and Supplementary Table 1). In this strain, glycerol cannot be converted to F6P, and therefore the strain cannot use glycerol alone to produce E4P, ribose 5-phosphate (R5P) and glucose 6-phosphate (G6P)[18]. Because these metabolites are essential, this strain becomes dependent on the activities of modules M1, M2 and M3 for growth. Although the enzymes of module M4, the C6-sugar cleavage module, are present, the strain does not rely on it for growth, as shown in Supplementary Fig. 7b as an exemplified flux distribution calculated with flux balance analysis (see Methods). We further deleted the *mgsA* gene, which encodes methylglyoxal synthase, to eliminate a potential sink lowering DHAP pool[49].

When we transformed the ΔFBP/GlpX strain with plasmids pFEIIO or pREIIO, respectively (Supplementary Table 1), we did not observe growth under selective conditions (i.e., on glycerol and sarcosine). Only when xylose was additionally supplemented, the strains grew (gray and orange lines in Fig. 4b and Supplementary Fig. 8). We reasoned that this was caused by the fact that (i) the ΔFBP/GlpX selection (in total 17% divided from F6P devotes 1% carbons in biomass, G6P 3%, R5P 10% and E4P 3%[50]) requires 6 times higher flux via modules M1 + M2 than the ΔfrmTKT selection (3%), and that (ii) the unique entry of E4P into the pentose phosphate pathway required *E. coli* to operate the pentose phosphate pathway in an unconventional manner.

Therefore, we started an adaptive laboratory evolution (ALE) experiment to alleviate and remove xylose-dependency of theses strains. At the start of the ALE experiment, we determined that 1 mM xylose was growth-limiting, restricting growth to an OD600 around 0.4 (orange line in Fig. 4b and Supplementary Fig. 8). From the same parent ΔFBP/GlpX strain containing plasmid pFEIIO, we started two

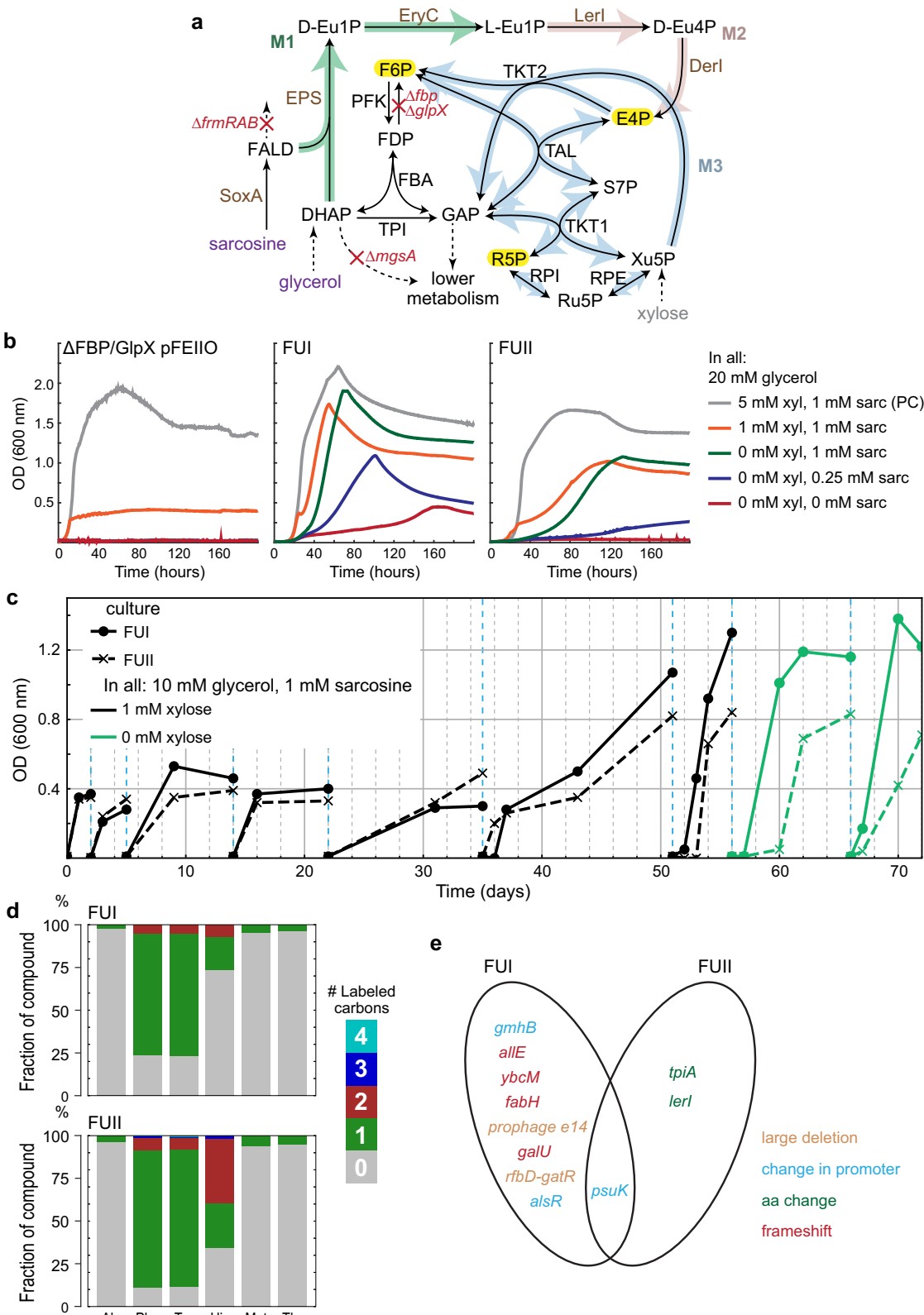

**Fig. 4 | ALE enables cooperation of modules M1, M2 and M3. a** A schematic representation of the ΔFBP/GlpX selection. Gene deletions are in red. Such a strain cannot synthesize the essential metabolites E4P, R5P and F6P, highlighted in yellow. Enzymes expressed from plasmid are in brown. Carbon sources are shown in purple. Abbreviations are the same as Fig. 3. **b** The growth behaviors of the unevolved parent strain and the evolved strains. xyl xylose, sarc sarcosine, PC positive control. **c** Adaptive laboratory evolution process of ΔFBP/GlpX pFEIIO. Two independent cultures were cultivated at xylose limiting condition and eventually eliminated xylose requirement for growth. **d** Labeling pattern of proteinogenic alanine (Ala), phenylalanine (Phe), tyrosine (Tyr), histidine (His), methionine (Met) and threonine (Thr), within the evolved strains FUI and FUII, upon feeding with unlabeled glycerol and sarcosine-(methyl-¹³C). The accompanied results with unlabeled sarcosine are shown in Supplementary Fig. 6b. **e** Mutations emerged from ALE within FUI and FUII. The mutations are detailed in Supplementary Data 1. Source data are provided as a Source Data file.

independent cultures simultaneously with 1 mM xylose in the medium (FUI/FUII). We re-inoculated the cultures into fresh medium after a few days of cultivation at a starting $OD_{600}$ of 0.01. Figure 4c shows the ALE process. After five passages, both cultures, FUI and FUII, were able to reach higher $OD_{600}$ and became able to grow on medium without xylose. We isolated five colonies for further characterization (Supplementary Table 1).

The five isolates were indeed confirmed to grow on xylose-free conditions. We observed identical growth between the three colonies of FUI and between the two colonies of FUII (Supplementary Table 1). While the FUI strains interestingly showed some sarcosine-independent growth (red line in Fig. 4b), sarcosine addition clearly increased growth (Fig. 4b). Overall, these results indicated that evolution had enabled the EuMP to sustain growth under the higher selection regime of ΔFBP/GlpX, and more importantly, the successful cooperation of the modules M1 to M3.

We further subjected the strains for [13]C labeling experiments and analyzed the [13]C labeling pattern of alanine, phenylalanine, tyrosine, histidine, methionine and threonine. Upon feeding unlabeled glycerol and N-methyl-[13]C labeled sarcosine, the labeling patterns of alanine, phenylalanine and tyrosine were similar to the ΔfrmTKT strains (see above), non-label in alanine and single-label in phenylalanine and tyrosine (Fig. 4d). However, the FUI strain showed a relatively high fraction of non-labeled aromatic amino acids that was different to the pattern of the FUII strain and higher than expected from natural isotopic distribution (Fig. 4d and Supplementary Fig. 6b). Similar to alanine, threonine and methionine were completely unlabeled, suggesting that label was neither introduced through aspartate (threonine and methionine backbone), nor C1-tetrahydrofolate (methionine methyl group). Finally, as observed for the aromatic amino acids, the labeling pattern of histidine differed between FUI and FUII, indicating some more complex metabolic scenarios. Only 19% of the histidine was single-labeled and 7% was double-labeled in FUI, while 26% and 38% single- and double-label, respectively, were observed in the FUII strain (Fig. 4d). This data confirmed that the EuMP was active in FUI and FUII, but also suggested distinct metabolic adaptations between the two different evolutionary lines.

## Mutation characterizations of evolved strains

To analyze the molecular basis of our ALE efforts and understand the differences between the two independent evolutionary lines, we sequenced whole genomes (including plasmids) of the evolved FUI and FUII strains via Illumina sequencing. While the different isolates within the FUI or FUII lines were genetically almost identical (Supplementary Data 1), the FUI and FUII lines accumulated distinct mutations. As shown in Fig. 4e and detailed in Supplementary Data 1, there was only one common mutation in both FUI and FUII strains, a nucleotide transversion in the promoter region of psuK gene. The psuK gene encodes pseudouridine kinase for the degradation of pseudouridine, which is the most abundant modified nucleoside in RNAs[51]. This promoter mutation was most likely involved in adapting the ΔFBP/GlpX strain via regulating psuK expression. FUI carried additional mutations, including (1) large deletions of the prophage e14 region and a 49 genes region ranging from rfbD to gatR; (2) frameshifts in genes of allE, ycbM, fabH, and galU; as well as (3) point mutations in promoter regions of gmhB and alsR. In the FUII strain, only two more missense mutations, one in tpiA and one in lerI, could be identified.

These results suggested that the FUI and FUII strains evolved different metabolic solutions under the same ΔFBP/GlpX selection regime, which is in line with the distinct tyrosine, phenylalanine and histidine labeling patterns between FUI and FUII. In FUI, the colanic acid (CA) cluster genes within the rfbD-garR region and galU, gmhB as well as fabH are involved in biosynthesis of nonessential extracellular polysaccharide and lipopolysaccharides (LPS) of the cell outer membrane[52–54]. Since F6P and G6P are precursors of CA and LPS[50,53], abovementioned null mutations relieved the requirements of F6P and G6P, thus lowering the selection pressure of the ΔFBP/GlpX strain. Note that Fbp and GlpX are the two main proteins carrying fructose-1,6-bisphosphatase (FBP) activity in E. coli, evidenced by that the ΔFBP/GlpX strain required xylose for growth (Fig. 4b and Supplementary Fig. 8). However, E. coli has additional enzymes with FBP activity: a third fructose-1,6-bisphosphatase YggF[55], haloacid dehalogenase (HAD)-like hydrolases YbhA and GmhB[54,56] as well as alkaline phosphatase (PhoA)[57]. These enzymes are able to substitute the conversion of fructose-1,6-bisphosphate (FDP) to F6P.

We speculated that in the evolved FUI strain, with lowered F6P requirements, these enzymes were able to produce sufficient F6P, which was further transformed via the reversible TKT2 reaction into E4P and Xu5P. Such internal E4P source is in line with the slight growth of FUI in the absence of xylose and sarcosine, as well as the better growth of FUI than FUII (Fig. 4b). Notably, it also explains the [13]C-labeling pattern in FUI, as the internally produced E4P is unlabeled, diluting the EuMP derived [13]C-labeled E4P in phenylalanine and tyrosine, as well as in histidine in FUI. For histidine, which was only 19% labeled once and 7% labeled twice, the net direction of TKT2, RPI and RPE reactions operated in reverse, so that a major portion of R5P was from Xu5P and a small portion was from TAL and TKT1 reactions (Fig. 5a). The observed mutation in the alsR promoter may have supported this scenario by altering the expression of rpiB. AlsR is a transcriptional repressor of rpiB[58], the encoded RpiB is the second ribose-5-phosphate isomerase (RPI) and potentially has DerI activity[32].

In FUII, mutated proteins TpiA and LerI are both EuMP pathway enzymes. These proteins are homologs and may possess activity of each other[32]. Thus, these mutations may have improved the EuMP flux in FUII. Note that in contrast to FUI, cell envelope biosynthesis was still intact in FUII, so that the F6P and G6P requirements were still high. This apparently precluded reverse operation of TKT2, as evidenced by classical labeling of phenylalanine and tyrosine. Regarding the labeling of histidine, we speculate that both the SBP and the TAL variants operated in FUII (Supplementary Fig. 1a and b). Although the strain does not contain GlpX, an identified sedoheptulose 1,7-bisphosphatase[21], other phosphatases may possess such activity, similar to the aforementioned FBP. Through the SBP variant (Fig. 5b), unlabeled and single-labeled histidine were derived from R5P and Xu5P, while the double labeled histidine originated from the TAL variant (Fig. 5c).

Indeed, the importance of both LerI and TpiA mutations was confirmed. Plasmid curing then re-transforming the plasmid pFEIIO (with naïve LerI, Supplementary Table 1) abolished growth of FUII strain (Supplementary Fig. 9a). FUII strain reverted the tpiA mutation to wild-type (FUII-tpiA_wt, Supplementary Table 1) did not grow in the presence of the plasmid with a mutated LerI (pFEIIOe, Supplementary Table 1 and Supplementary Fig. 9b). Although introducing a mutation of either LerI or TpiA alone was not sufficient to enable the growth of the parental ΔFBP/GlpX strain, together, they allowed growth of the strain (Supplementary Fig. 9). These results indicate synergy between mutations in LerI and TpiA. Finally, considering the regulation role of PsuK (as mentioned above), the extended lag phase observed in the reverse-engineered ΔFBP/GlpX-tpiA_m strain with pFEIIOe, compared to the FUII strain, may suggest an additional layer of synergy with psuK promoter mutation (Supplementary Fig. 9b and Fig. 4b).

## Discussion

One-carbon compounds play a central role in a circular bioeconomy. Formaldehyde is an important intermediate in several natural and synthetic C1-assimilation pathways. In this study, we present a new-to-nature formaldehyde assimilation pathway, the EuMP cycle, following a mix-and-match engineering strategy similar to the homoserine cycle[23]. Our theoretical calculations highlight that the EuMP cycle is equally efficient as the most energy efficient natural RuMP cycle,

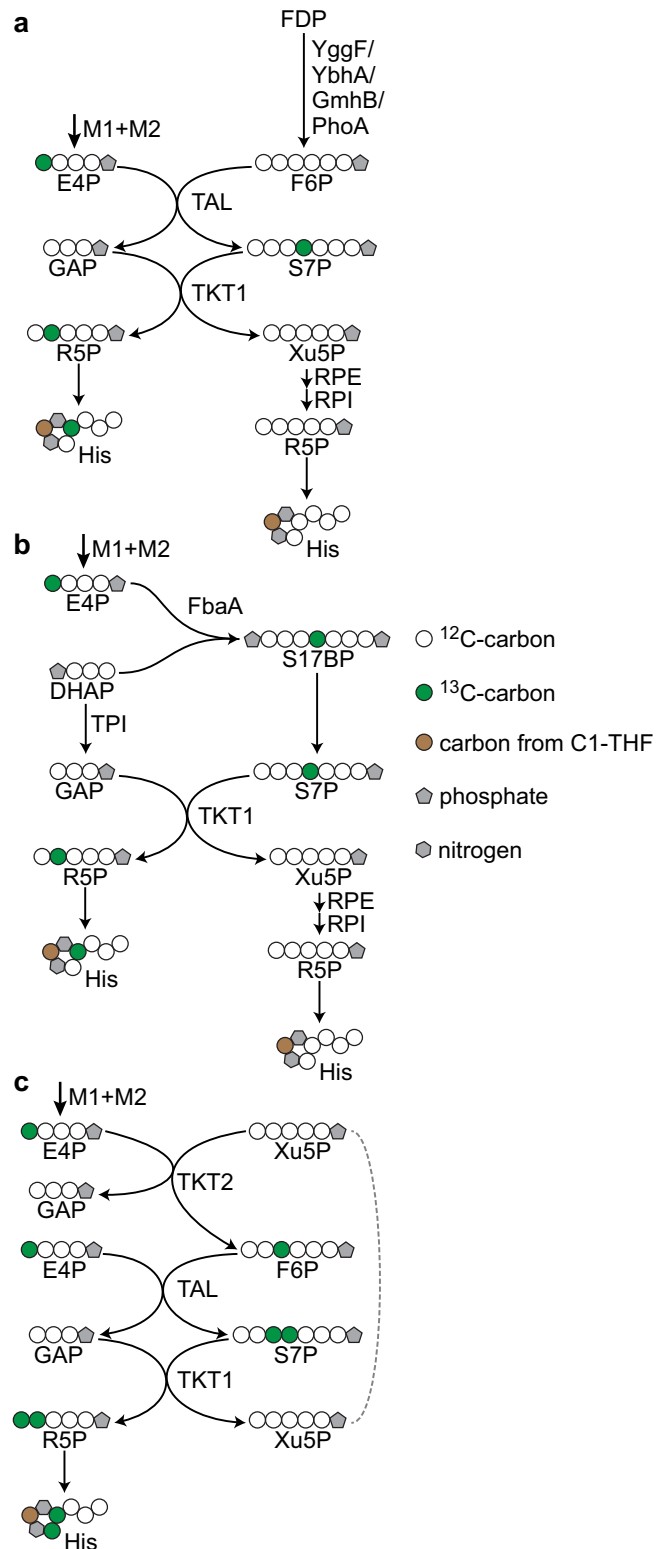

**Fig. 5 | Possible histidine labeling traces of FUI and FUII strains.** From [13]C-labeled formaldehyde and unlabeled DHAP, fructose-1,6-bisphosphatase activity from YggF, YbhA, GmhB, or PhoA generates unlabeled F6P, resulting in histidine (His) unlabeled and once labeled (**a**). Histidine would be in unlabeled and once labeled when EuMP operates via the sedoheptulose 1,7-bisphosphatase (SBP) route (**b**). And it would be twice [13]C-labeled when EuMP operates in the transaldolase (TAL) manner (**c**).

outperforming other natural C1-assimilation pathways. Although the EuMP cycle requires more enzymes than the RuMP cycle, it is hard to compare the enzyme costs of both pathways because of the difficulties of obtaining full kinetics of the core enzymes. Still, the EuMP cycle serves as a complementation to the synthetic homoserine cycle, which has relatively low theoretical yields in biomass and sugar phosphates[23]. EuMP can operate in four different modes, offering metabolic flexibility and potentially enhancing adaptability to various requirements. Moreover, the EuMP cycle holds the potential to be further extended by phosphoketolase reaction to further improve its efficiency, as phosphoketolase produces C2 moieties from sugar phosphates without oxidation (releasing $CO_2$)[59].

In designated *E. coli* selection strains, we demonstrated the activity of the core modules of the EuMP cycle. While we established EPS activity through FucA and RhaD from *E. coli*, enzyme engineering or screening of additional candidates, such as those from *Propionibacterium pentosaceum*[60] and Swiss chard leaves[38], might provide even more specific and/or active enzymes in the future. An L-Eu1P producing EPS would be especially interesting, as it omits the necessity of EryC overexpression. Such enzymes could also benefit other applications, for instance rare sugars production[61].

Similarly, better enzymes for the isomerases, i.e., EryC, LerI and DerI, can also be identified and substituted. Microorganisms catabolizing tetritols offers a broad mining source. It was suggested that EPS in *P. pentosaceum*[60] and *Sinorhizobium meliloti*[62] functions in the Eu1P cleavage direction for erythritol catabolism. Further study on the co-occurrence of EPS and these isomerases (as well as the other tetritol catabolism enzymes) is particularly interesting, because it could not only elucidate their physiological functions from evolutionary and ecological aspects, but may also find that the EuMP cycle is already a naturally existing C1-assimilation pathway, similar to the reductive glycine pathway[63].

The EuMP pathway assimilates formaldehyde with DHAP and enters central metabolism on the level of E4P. This unique feature of the pathway makes ALE necessary in our (and potentially also future) engineering efforts. Note that the pentose phosphate pathway operates usually from C5 or C6 sugars. E4P is typically "only" an intermediate and a branching point into different biosynthetic pathways, thus rather serving as sink than as source in the classical pentose phosphate pathway. Entering from E4P required rewiring the fluxes of the highly regulated pentose phosphate pathway. In ΔFBP/GlpX with increased selection pressure, ALE was used to establish EuMP dependent growth for the FucA-based overexpression strain. In the FUI strain nutrient requirements from EuMP were lowered by abolishing CA and LPS biosynthesis (a strategy is also used for cell factory optimization[53]), while in the FUII strain EuMP pathway activity was tuned via mutations in the pathway enzymes TpiA and LerI in a synergistic manner. In summary, two independent evolutionary lines allowed us to establish the first three modules of the EuMP cycle.

The in vivo establishment of the full EuMP cycle relying on formaldehyde solely will require more metabolic engineering and long-term ALE. Module M4 should be active itself as it is part of glycolysis and operates in the same direction as when feeding with glucose; integrating it with the other three modules is still not an easy task. The ΔFBP/GlpX selection strain would still be a good evolution platform, because it contains all four EuMP modules while relying on modules M1, M2 and M3 to grow. Mutations identified from FUI and FUII could be established in the strain. Blocking glycolysis would be beneficial, as it lowers the initial selection pressure[64] and avoids interference from sarcosine-derived glycine. Such a strain would be subjected to further ALE with decreasing glycerol and increasing sarcosine concentrations. Similar approaches have been used in this study and been applied

previously in successfully installing the RuMP cycle[17,19] and the Calvin–Benson–Bassham cycle[64,65] heterologously in *E. coli*. Although we observed partial bypass in the FUI ALE, similar occurrences restoring growth are unlikely in such a full cycle selection condition which has a selection pressure similar to the abovementioned examples. In silico modeling, such as flux balance analysis[19] and ensemble modeling for robustness analysis[17], may also be conducted to identify potential bottlenecks for rational strain engineering. These strategies could enable the growth of *E. coli* via the full EuMP cycle first on sarcosine-derived formaldehyde, decoupling the challenge of formaldehyde production from methanol, formate, or other C1 feedstocks, paving the way for the future development of growth solely on C1 feedstocks. While achieving the goal is undoubtedly challenging, it holds the promise as an efficient C1-platform in bioindustry.

## Methods

### Max-min driving force analysis

Max-min driving force (MDF) analysis[34] was applied to evaluate the thermodynamic feasibility of the EuMP cycle for glyceraldehyde 3-phosphate production from formaldehyde. Python packages equilibrator_api (v 0.4.7) and equilibrator_pathway (v 0.4.7) were used for the calculations. The changes in Gibbs energy of the reactions were estimated using the component contribution method[66]. Cofactor and metabolite concentrations were constrained to the range 1 μM to 10 mM[34] with a change for formaldehyde upper bound to 0.5 mM[18]. pH was assumed to be 7.5, ionic strength was assumed to be 0.25 M, and $-\log[Mg^{2+}]$ (pMg) was assumed to be 3. The scripts and details are available in the following open-access archive repository: https://github.com/he-hai/PubSuppl, within "2022_EuMP" the directory.

### Flux balance analysis

Flux balance analysis modeling was conducted with COBRApy using the most updated *E. coli* genome-scale metabolic model iML1515[67]. The model was modified as follows: (i) transhydrogenase (THD2pp) translocates one proton instead of two[68]; (ii) homoserine dehydrogenase (HSDy) was set to irreversibly produce homoserine[23]; (iii) anaerobic relevant reactions, PFL, OBTFL, FDR2, and FDR3, were removed from the model; (iv) POR5, GLYCK, FDH4pp FDH5pp, GART, DRPA, and PAI2Twere also knocked out to block unrealistic routes; (v) the ATP maintenance reaction (ATPM) was also switched off.

To compare the yields of the 12 building block precursors[35] and biomass from formaldehyde assimilation via EuMP and the three natural pathways, the models of the four pathways were integrated in the modified *E. coli* model. The maximum yields were estimated using formaldehyde as feedstock.

To showcase the flux distributions of the selection strains, gene knock-outs were stimulated within the EuMP *E. coli* model. Bounds of the biomass reaction were fixed to be 1 and the objective function was changed to minimal the EPS flux.

### Strains and genomic manipulation

All strains used in the study are listed in Supplementary Table 1. An *E. coli* K-12 MG1655 derived SIJ488 strain[69] was used as host platform. Gene knock-outs were obtained by λ-Red recombination or P1 transduction[31]. Recombineering knock-out cassettes were generated by PCR amplifying the FRT-PGK-gb2-neo-FRT (Km) cassette (Gene Bridges, Germany) using 50 bp homologous arms from the target genes in the primers (Supplementary Data 2). About 300 ng of cassette DNA was transformed into freshly prepared electrocompetent cells, which were induced 45 min prior with 15 mM L-arabinose when OD$_{600}$ was around 0.3. The knock out colony was selected on LB kanamycin plates and verified by colony PCR with "Ver" primers (Supplementary Data 2). Flippase was induced by 50 mM L-rhamnose to remove antibiotic markers. And its removal was confirmed by colony PCR.

CRISPR/Cas9 system was used to reengineer the mutation (G94D) of TpiA into the parent strain (ΔFBP/GlpX, Supplementary Table 1) following the Scar-less method[70] using a donor oligo (tpiA_m, Supplementary Data 2). The mutation of *tpiA* in resulting strain, ΔFBP/GlpX-tpiA_m (Supplementary Table 1), was confirmed by Sanger sequencing. To revert the mutation of TpiA in FUII, a chloramphenicol selection marker (Cm) was first inserted downstream of the *tpiA* gene within the parent strain (ΔFBP/GlpX, Supplementary Table 1) using the λ-Red system mentioned above. P1 phage transduction[71] was then conducted transferring the wild type *tpiA* to FUII2 strain using the resulting strain (tpiA::Cm, Supplementary Table 1) as donor. Sequencing confirmed the reversion of *tpiA* gene.

### Plasmid construction

All plasmids are listed in Supplementary Table 1 and their constructions were followed the method in ref. 31. Heterologous gene sequences were listed in Supplementary Data 3. *fucA*, *rhaD*, *fbaA* and *yihT* were cloned from *E. coli* MG1655 genome with primers listed in Supplementary Data 2. All genes were inserted into pNivC plasmid individually under a ribosome binding site "C" (AAGTTAAGAGGCAAGA). Apart from *lerK* and *eltD*, all others have a 6xHis-tag right after the start codon. Genes were then assembled into an operon via BioBrick enzymes: *BcuI*, *SalI*, *NheI* and *XhoI* (FastDigest, Thermo Scientific). *EcoRI* and *PstI* (FastDigest, Thermo Scientific) were used to move the whole operon to an expression vector (pZ plasmid), which has a synthetic promoter pgi-20 or pgi-10, p15A medium copy origin as well as streptomycin selection marker.

### Growth media and growth experiments

LB (1% NaCl, 0.5% yeast extract, 1% tryptone) was used in cultivating cells for all gene manipulation e.g., gene cloning and gene knocking out. Antibiotics were used at concentrations of 50 μg/mL kanamycin, 100 μg/mL ampicillin 100 μg/mL streptomycin, or 30 μg/mL chloramphenicol to the medium if needed. M9 minimal medium (Sigma-Aldrich, Germany) supplemented with 2 mM MgSO$_4$, 100 μM CaCl$_2$ and trace elements (134 μM EDTA, 31 μM FeCl$_3$, 6.2 μM ZnCl$_2$, 0.76 μM CuCl$_2$, 0.42 μM CoCl$_2$, 1.62 μM H$_3$BO$_3$, 0.081 μM MnCl$_2$) was used for growth experiments. Carbon sources used were indicated in the main text. For ΔtktAB and ΔfrmTKT strains, E4P supplements (E4P_suppl.) were also added in relaxing medium at final concentration of: 1 mM shikimic acid, 1 μM pyridoxine, 250 μM tyrosine, 500 μM phenylalanine, 200 μM tryptophan, 6 μM 4-aminobenzoic acid, 6 μM 4-hydroxybenzoic acid and 50 μM 2,3-dihydroxybenzoic acid[18].

Growth experiments started with preculturing the cells in 4 mL M9 medium in glass tubes under relaxing conditions. When reaching late exponential phase, the precultures were washed three times with sterilized Mili-Q water by centrifugation and resuspension. Cells were inoculated in dedicated testing conditions in 96-well plates (Nunclon Delta Surface, Thermo Scientific) at starting OD$_{600}$ of 0.01 cultured at 37 °C in microplate reader (BioTek EPOCH 2). Each well contained 150 μL culture and covered with 50 μL sterilized mineral oil (Sigma-Aldrich, Germany). The shaking program cycle (controlled by Gen5 v3) has 4 shaking phases, lasting 60 s each: linear shaking followed by orbital shaking, both at an amplitude of 3 mm, then linear shaking followed by orbital shaking both at an amplitude of 2 mm. The optical density (OD$_{600}$) in each well was monitored and recorded after every three shaking cycles (~16.5 min). Raw data from the plate reader were calibrated to normal cuvette measured OD$_{600}$ values according to Eq. 1.

$$OD_{cuvette} = OD_{plate} / 0.23 \qquad (1)$$

MATLAB (MathWorks) was used for calculating growth parameters based on three technical triplicates – the average values were

used to generate the growth curves. Checked in MATLAB, in all cases variability between triplicates measurements were less than 5%.

## Molecular phylogenetic analysis

DHAP aldolase sequences from *E. coli*, FucA P0AB87, RhaD P32169, YihT P32141, FbaA P0AB71, GatY P0C8J6, KbaY P0AB74 and FbaB P0A991, were obtained from UniProt. MAFFT v7.490 was used for multiple sequence alignment with default parameters. MEGA 10.0.5 used the aligned sequences to construct a Neighbor-joining phylogenetic tree, with 1000 Bootstrap replication tests.

## Stable isotopic labeling

Sarcosine-(methyl-$^{13}$C) was purchased from Sigma-Aldrich. The strains were cultivated in glass tubes on M9 minimal media with appropriate carbon sources. Cells were harvested at the late exponential phase. The equivalent volume of 1 mL of culture at $OD_{600}$ of 1 was harvested and washed by centrifugation. Experiments were performed with replicates which in all cases showed identical results ($\pm$ 5%). Proteinogenic amino acids were hydrolyzed from biomass with 6 M HCl at 95 °C for 24 h[72]. The samples were completely dried under a stream of $N_2$ at 95 °C. Hydrolyzed amino acids were analyzed with UPLC–ESI–MS with a Waters Acquity UPLC system (Waters), using an HSS T3 $C_{18}$ reversed phase column (100 mm × 2.1 mm, 1.8 µm; Waters)[23]. 0.1% formic acid in $H_2O$ (A) and 0.1% formic acid in acetonitrile (B) were the mobile phases. The flow rate was 0.4 mL/min and the gradient was: 0 to 1 min – 99% A; 1 to 5 min – linear gradient from 99% A to 82%; 5 to 6 min – linear gradient from 82% A to 1% A; 6 to 8 min – kept at 1% A; 8-8.5 min—linear gradient to 99% A; 8.5-11 min – re-equilibrate. Mass spectra were acquired using an Exactive mass spectrometer (Thermo Scientific) in positive ionization mode, with a scan range of 50.0 to 300.0 m/z. The spectra were recorded during the first 5 min of the LC gradients. Data analysis was performed using Xcalibur (Thermo Scientific). Determination of retention times was performed by analyzing amino-acid standards (Sigma-Aldrich) under the same conditions.

## Adaptive laboratory evolution

To evolve ΔFBP/GlpX pFEIIO strains (Supplementary Table 1) for formaldehyde dependent growth, xylose concentration was tested firstly. The resulting limiting condition of 1 mM xylose was used as a starting point. The strains were first inoculated from cyro-glycerol stocks onto LB streptomycin plates, then inoculated to M9 liquid medium with 10 mM glycerol, 1 mM xylose and 1 mM sarcosine in two glass tubes of each strain. The two independent cultures were subcultured to fresh media at initial $OD_{600}$ of 0.01. After seven passages, both pFEIIO cultures were then cultivated at no xylose conditions for another two passages. The cultures were then streaked out onto agar plates and single colonies were isolated (Supplementary Table 1).

## Whole genome sequencing and analysis

NucleoSpin Microbial DNA kit (MACHERY-NAGEL, Düren, Germany) was used for genomic DNA extraction. Library construction and genome sequencing were performed by Novogene (Cambridge, United Kingdom) using the paired-end Illumina sequencing platform. Analysis of the sequencing data was carried out using *breseq* (v0.31.0)[73] with MG1655 (GenBank: NC_000913) and pFEIIO as references.

## Reporting summary

Further information on research design is available in the Nature Portfolio Reporting Summary linked to this article.

## Data availability

Raw reads of NGS are deposited at NCBI and can be accessed under BioProject PRJNA895983. MG1655 genome is available at Genbank under accession NC_000913. Protein sequences of DHAP aldolases

from *E. coli* are obtained from UniProt: FucA P0AB87, RhaD P32169, YihT P32141, FbaA P0AB71, GatY P0C8J6, KbaY P0AB74, and FbaB P0A991. Source data are provided with this paper.

## Code availability

Full code and models of MDF and flux balance analysis are available at Github [https://github.com/he-hai/PubSuppl] within the "2022_EuMP" directory. They can also be found under https://doi.org/10.5281/zenodo.10207936[74].

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

## Acknowledgements

We thank Änne Michaelis for experimental assistance. We thank Shanshan Luo for critical reading of the manuscript. This study was funded by the Max Planck Society. Tong Wu and Hai He were also funded by the China Scholarship Council.

## Author contributions
Conceptualization, P.M., A.B.-E. and H.H.; Methodology, T.W., P.A.G.C, A.K. and H.H.; Software, H.H.; Validation, T.W, P.A.G.C and A.K.; Formal analysis, T.W, P.A.G.C, A.K. and H.H.; Investigation, T.W, P.A.G.C and A.K.; Resources, T.J.E. and A.B.-E.; Writing – Original Draft, H.H.; Writing—Review & Editing, T.W., P.A.G.C., S.L., P.M., T.J.E. and H.H.; Visualization, H.H.; Supervision, A.B.-E. and H.H.; Funding Acquisition, T.J.E. and A.B.-E.

## Funding

## Competing interests
The authors declare no competing interests.
