## [Peer Review File · Nature Communications]

Engineering a synthetic energy-efficient formaldehyde assimilation cycle in *Escherichia coli*Reviewers' Comments:

Reviewer #1:

Remarks to the Author:

o General comments:

In this study, the authors established an Erythrulose MonoPhosphate cycle in model host *E. coli* to efficiently assimilate formaldehyde. In silico validation of the EuMP pathway showed the same energy demands and comparable intermediate yields between synthetic EuMP and one of the most efficient formaldehyde assimilation pathways in nature, RuMP. Four modules of EuMP coupled with the SoxA gene allowed formaldehyde from sarcosine to condense with DHAP to generate GAP. To enable this novel formaldehyde assimilating pathway in vivo, a high-selection-pressure mutated *E. coli* strain was used as a background strain and an adaptive evolution approach was employed. The authors demonstrated the incorporation of formaldehyde via the EuMP pathway via an isotopic labelling approach. Ultimately, the whole genomes of the final evolved strains were sequenced and analyzed. The authors have generally provided a well-writing manuscript with fully supporting data. The finding of a new efficient EuMP cycle can contribute to developing a strong synthetic methylotrophic strain. Nevertheless, the experimental flow and concept of this study are similar to a previous report (Ref: He, H., Höper, R., Dodenhöft, M., Marlière, P., & Bar-Even, A. (2020). An optimized methanol assimilation pathway relying on promiscuous formaldehyde-condensing aldolases in *E. coli*. *Metabolic engineering*, 60, 1-13). Because of this, it would be preferable if the authors could expand on the outstanding of this work in comparison to the prior one. Below are some specific comments on this manuscript.

o Major comments:

Line 185-186: The author stated that "In these strains, growth rates and maximal ODs directly correlated with increasing sarcosine concentrations (Fig. 3 c and d), while no growth was observed without sarcosine and growth could not be restored without EPS (i.e., *fucA* or *rhaD*) overexpression (Fig. 3e)". However, when looking into the Figure 3c and 3d, without sarcosine (0 mM, red line), the pFEIIO and pREIIO strains seemed to grow though slowly. Please explain this.

From the literature, a formaldehyde assimilation reaction relying on serine aldolase has been constructed to condense glycine and formaldehyde, followed by the homoserine cycle (Ref: He, H., Höper, R., Dodenhöft, M., Marlière, P., & Bar-Even, A. (2020). An optimized methanol assimilation pathway relying on promiscuous formaldehyde-condensing aldolases in *E. coli*. *Metabolic engineering*, 60, 1-13). The homoserine cycle and the newly developed EuMP pathway are equal in terms of energy cost. What is the appealing point of the erythrulose 1-phosphate formaldehyde-lyase developed in this study compared with serine aldolase from the work mentioned above? The authors should show this point in the discussion section.

In this study, the author used sarcosine as a formaldehyde donor. It is somewhat not suitable with the originality of this study: methanol/formate are potential one-carbon feedstocks for biotechnology. I wonder if the EuMP cycle can work while using methanol (or formate) as a substrate.

o Minor comments:

Please provide figures with high resolution.

Line 128: 'In' should be in lowercase form 'in'

Line 163: Please add a space between 'dEu1P' and 'when'

Line 227: 'grey and orange lines' (not 'green lines'), I guess?

Reviewer #2:

Remarks to the Author:

This is an excellent research paper demonstrating the feasibility of a new formaldehyde assimilatory cycle in *E. coli*. The *in vivo* operation of this pathway is fully demonstrated. The reading is enjoyable and results are of very high scientific quality, original and convincing. Comments that would make the article even more impactful are mentioned below.

Major:

Point 1: The computational approach showed that EuMP cycle is equally efficient as the RuMP however it requires the expression of twice more genes to be functional in *E. coli*. Only *hps* and *phi* are required to construct a RuMP cycle starting from formaldehyde in *E. coli* versus four genes (i.e. *EPS*, *EryC*, *LerI* and *DerI*) for the EuMP cycle. I thus wonder what are the advantages of such pathway. Is a higher growth rate expected on methanol compare to the RuMP based synthetic methylotrophs recently obtained by Chen et al., 2020 and Kim et al., 2020 and Keller et al., 2022? This should be discussed in the revised manuscript.

Point 2: In the revised manuscript, the authors should better explain their choice on the origin of the genes encoding for the enzymes of the M2 module as it has been done for the *EPS* encoding genes. Indeed, authors choose to express the genes from *M. smegmatis* but those enzymes are also found in others bacteria (e.g. *B. abortus*) as mentioned in BioCyc. Did the authors tried other genes encoding enzymes?

Point 3: Figure 2 shows that the plasmid pKIID rescue growth of the Δ tktAB strain on 10 mM L-threitol and erythritol, but the growth starts after a 40 to 50 hours of lag phase. How can it be explained? What happens during this long lag phase?

Point 4: L185: Based on the figure 3, authors claim that no growth was observed without sarcosine in the strains overexpressing pFEIIO or PFEIIO plasmid but in these conditions (red line figure 3c&d) a small growth is observed. How can it be explained?

Point 5: In figures 3 h and 4 d, there is always few amounts of labelling that is found in Alanine although no labelled should be found as is it claimed in L212 and 259. The same comments can be done for the labelling pattern of tyrosine and phenylalanine which are supposed to be single-labeled as claimed in L 208 and 259 but where few double-labeled is observed. How can it be explained?

Point 6: L341: authors said that: "The *in vivo* establishment of the full EuMP cycle relying on formaldehyde solely will require more metabolic engineering and long-term ALE. However, short ALE experiments performed in this work demonstrated that *E. coli* was able to partially bypass the EuMP cycle after 5 passages of evolution. Isn't the cycle likely to be completely hijacked with longer ALE experiments? This point should be discussed in the revised manuscript.

Minor:

L100: "and NAD(P)H" can be removed as the mentioned cycles do not use it.

Fig 1. The *EryC* reaction should be added in the schematic representation of the EuMP cycle.

Figure 3 g: unlabeled PEP can also derive from the GAP originating from TAL reaction. This should be added.

Reviewer #3:

Remarks to the Author:

He et al. devised a new-to-nature formaldehyde fixation pathway and embedded it into *E. coli* for

efficient C1 assimilation. This pathway synthetic formaldehyde assimilation pathway was named the erythrose monophosphate (EuMP) cycle and it relies on the side activity of dihydroxyacetone phosphate (DHAP) dependent aldolase. This erythrose 1-phosphate synthase reaction condenses formaldehyde with DHAP to yield L-erythrose 1-phosphate. By two isomerase reactions, the latter can be converted to D-erythrose 4-phosphate (E4P), an intermediate of the pentose phosphate shunt. Conversion of E4P by reactions of the pentose phosphate shunt, triosephosphate isomerase, phosphofructokinase and fructose biphosphate aldolase regenerates DHAP. In addition, three further variants involving sedoheptulose 1,7-bisphosphate adolase and/or the Entner-Doudoroff variant of glycolysis. However, the chosen transaldolase/fructose biphosphate aldolase is the most energy efficient variants.

Embedding into *E. coli* metabolism in vivo, was followed by gradual implementation of 4 modules. First, E4P was isolated by deletion of transketolase genes and a mutant expressing genes of tetrytol catabolism provided E4P from either erythritol or threitol. This demonstrated that isomerase LerI and DerI provided sufficient E4P for growth of *E. coli* (module 2). Next, erythrose 1-phosphate dependent formaldehyde fixation was established. Side activities of a number of aldolase regarding the aldehyde acceptor substrate (but not the donor substrate) were sought in literature for the aldolases depending on DHAP as donor. Formaldehyde as acceptor substrate was generated from sarcosine by sarcosine oxidase (in situ, adjustable in a strain lacking formaldehyde oxidation to formate). Of several candidates, L-fuculose 1-phosphate aldolase (FucA) from *E. coli* performed best to assimilate formaldehyde (module 1). Tracer experiments support this notion. Not surprisingly, the native pentose phsosphate shunt enzymes did not operate to allow for efficient integration of the modules 1 and 2 and made adaptive laboratory evolution (ALE) necessary. First, a double deletion lacking both fructose bisphosphatases was generated which cannot grow with glycerol as sole carbon source as E4P, ribose 5-phosphate and glucose 6-phosphate cannot be generated. However, when modules 1 and 2 operate, these intermediates can be generated from sarcosine and glycerol. Two independent ALE mutants were selected and grew well with glycerol and sarcosine. Tracer experiments support this notion. Genome sequencing of the ALE mutatns showed that these shared only one mutation: a SNP in the promoter region of *psuK* that codes for pseudouridine kinase, an enzyme required for degradation of pseudouridine, a nucleoside found, e.g., in tRNAs. It remains unclear if and how the promoter activity of *psuK* was affected or if this SNP is at all relevant for the selected growth phenotypes of ALE mutants FUI and FUII.

In FUII, mutated proteins TpiA and LerI provided a likely explanation for the observed phenotype. In FUI, the requirement for fructose 6-phosphate was lowered. Would introduction of the SNPs in the TpiA and LerI genes in FUII improve its growth with glycerol and sarcosine?

The authors do not discuss module 4 much: why?

The enzymes with erythrose 1-phosphate synthase side activity were important to establish this new-to-nature cycle. Is it actually new-to-nature? May be the donor microorganisms operate this or a related cycle for formaldehyde assimilation. Do these enzymes occur in combination with enzymes of tetrytol catabolism? Sarcosine oxidase? Can we learn from evolutionary considerations and absence/occurrence of these side activities?

In prior experiments, formaldehyde was shown to react chemically with DNA and the appropriate compensatory mutations were selected. Here, these genes were not mutated. Is this due to the sarcosine strategy of providing formaldehyde?

I. 104: FBA and FBA: enzyme or flux balance analysis?

Fig. 2: the abbreviation d is often sued for desoxy: why not rename dEu4P to DEu4P or D-Eu4P (similar lEu1P to L-Eu1P)?

I.134 what is E4P_suppl.?

Reviewer #1 (Remarks to the Author):

General comments:

In this study, the authors established an Erythrulose MonoPhosphate cycle in model host *E. coli* to efficiently assimilate formaldehyde. In silico validation of the EuMP pathway showed the same energy demands and comparable intermediate yields between synthetic EuMP and one of the most efficient formaldehyde assimilation pathways in nature, RuMP. Four modules of EuMP coupled with the SoxA gene allowed formaldehyde from sarcosine to condense with DHAP to generate GAP. To enable this novel formaldehyde assimilating pathway in vivo, a high-selection-pressure mutated *E. coli* strain was used as a background strain and an adaptive evolution approach was employed. The authors demonstrated the incorporation of formaldehyde via the EuMP pathway via an isotopic labelling approach. Ultimately, the whole genomes of the final evolved strains were sequenced and analyzed. The authors have generally provided a well-writing manuscript with fully supporting data. The finding of a new efficient EuMP cycle can contribute to developing a strong synthetic methylotrophic strain. Nevertheless, the experimental flow and concept of this study are similar to a previous report (Ref: He, H., Höper, R., Dodenhöft, M., Marlière, P., & Bar-Even, A. (2020). An optimized methanol assimilation pathway relying on promiscuous formaldehyde-condensing aldolases in *E. coli*. *Metabolic engineering*, 60, 1-13). Because of this, it would be preferable if the authors could expand on the outstanding of this work in comparison to the prior one. Below are some specific comments on this manuscript.

We thank the reviewer for the nice words and support. We revised the text accordingly in the discussion, adding similarities and advantages over the homoserine work:

“In this study, we present a new-to-nature formaldehyde assimilation pathway, the EuMP cycle, following a mix-and-match engineering strategy similar to the homoserine cycle ... It serves as a complementation to the synthetic homoserine cycle, which has relatively low theoretical yields in biomass and sugar phosphates. Moreover, the EuMP cycle holds the potential to be further extended by phosphoketolase reaction to further improve its efficiency, as phosphoketolase produces C2 moieties from sugar phosphates without oxidation (releasing CO₂)”

Major comments:

Line 185-186: The author stated that “In these strains, growth rates and maximal ODs directly correlated with increasing sarcosine concentrations (Fig. 3 c and d), while no growth was observed without sarcosine and growth could not be restored without EPS (i.e., *fucA* or *rhaD*) overexpression (Fig. 3e)”. However, when looking into the Figure 3c and 3d, without sarcosine (0 mM, red line), the pFEIIO and pREIIO strains seemed to grow though slowly. Please explain this.

We thank the reviewer for the comment. Indeed, we observed the phenomenon in multiple replicated experiments. However, only if the full pathway was overexpressed. Hence, we attribute it to the presence of internal source formaldehyde. Our hypothesis can be supported by evidences: (1) formaldehyde is non-depletable from cellular metabolism, many biological processes produce this compound (Ref: 9 & 11); (2) such growth was still pathway enzymes dependent; (3) relatively too high fractions of unlabeled Phe and Tyr (Fig. 3h);

(4) both FucA and RhaD supported growth under a low sarcosine condition (0.2 mM, orange lines) and at the same time such growth was not observed in the higher selection Δ FBP/GlpX strain.

To also explain this to the readers, we revised the text:

“In these strains, growth rates and maximal ODs directly correlated with increasing sarcosine concentrations (Fig. 3 c and d) and growth could not be restored without EPS (i.e., *fucA* or *rhaD*) overexpression (Fig. 3e). ... We attributed the small growth under sarcosine absent condition (red lines in Fig. 3 c and d) to the non-depletable internal source formaldehyde, because such growth was FucA/RhaD and EryC dependent (Fig. 3 e, f and Supplementary Fig. S5).”

“... the relatively high unlabeled fractions, especially for FucA, were likely from internal source formaldehyde originated from unlabeled glycerol.”

From the literature, a formaldehyde assimilation reaction relying on serine aldolase has been constructed to condense glycine and formaldehyde, followed by the homoserine cycle (Ref: He, H., Höper, R., Dodenhöft, M., Marlière, P., & Bar-Even, A. (2020). An optimized methanol assimilation pathway relying on promiscuous formaldehyde-condensing aldolases in *E. coli*. *Metabolic engineering*, 60, 1-13). The homoserine cycle and the newly developed EuMP pathway are equal in terms of energy cost. What is the appealing point of the erythrulose 1-phosphate formaldehyde-lyase developed in this study compared with serine aldolase from the work mentioned above? The authors should show this point in the discussion section.

We thank the reviewer for the comment. The EuMP has equal energy cost as the RuMP, which is the most efficient natural pathway, but not the same energy cost as the homoserine cycle. The values shown in the homoserine cycle paper compare costs for acetyl-CoA production. In our case here, the product of EuMP is GAP. A comparison between the EuMP cycle and the homoserine cycle for bioproduction is the same to RuMP-homoserine. This is shown in the homoserine cycle paper Fig. 2. As shown by that figure and Fig 1c in the present study, our calculations indicated that the EuMP cycle supports the highest yields in biomass and for 11 out of the 12 precursors, the exception is acetyl-CoA. Therefore, we added in the discussion:

“It serves as a complementation to the synthetic homoserine cycle, which has relatively low theoretical yields in biomass and sugar phosphates.”

In this study, the author used sarcosine as a formaldehyde donor. It is somewhat not suitable with the originality of this study: methanol/formate are potential one-carbon feedstocks for biotechnology. I wonder if the EuMP cycle can work while using methanol (or formate) as a substrate.

The reviewer is correct, we have also described in the introduction, that methanol/formate are the favorable feedstocks. We argue that using sarcosine as a formaldehyde donor is an aid rather a drawback of our study, reasoning that a good formaldehyde production module of methanol oxidation or formate reduction is still under development. We did test methanol as a substrate within the Δ frmTKT selection at the beginning and got no

growth. Therefore, we decouple the challenges to demonstrate the new-to-nature pathway on the efficient formaldehyde source of sarcosine first, and then move to use methanol/formate as carbon source in the future. We revised the text:

“Sarcosine oxidase (SoxA), finally, was used for providing formaldehyde via a reliable method from sarcosine *in situ*. To focus on the demonstration of the EuMP pathway, we used this established formaldehyde production modular to set aside the inefficient production of formaldehyde from C1 substrates in this study.”

“...decoupling the challenge of formaldehyde production from C1 ...”

Minor comments:

Please provide figures with high resolution.

We apologize the low resolution of figures. We now have higher resolution figures embedded. We provide also vector files of the figures.

Line 128: ‘In’ should be in lowercase form ‘in’

Corrected.

Line 163: Please add a space between ‘dEu1P’ and ‘when’

Corrected.

Line 227: ‘grey and orange lines’ (not ‘green lines’), I guess?

We thank the reviewer for the correction.

Reviewer #2 (Remarks to the Author):

This is an excellent research paper demonstrating the feasibility of a new formaldehyde assimilatory cycle in *E. coli*. The *in vivo* operation of this pathway is fully demonstrated. The reading is enjoyable and results are of very high scientific quality, original and convincing. Comments that would make the article even more impactful are mentioned below.

We thank the reviewer for the kind words and support.

Major:

Point 1: The computational approach showed that EuMP cycle is equally efficient as the RuMP however it requires the expression of twice more genes to be functional in *E. coli*. Only hps and phi are required to construct a RuMP cycle starting from formaldehyde in *E. coli* versus four genes (i.e. EPS, EryC, LerI and DerI) for the EuMP cycle. I thus wonder what is the advantage of such pathway. Is a higher growth rate expected on methanol compare to the RuMP based synthetic methylotrophs recently obtained by Chen et al., 2020 and Kim et al., 2020 and Keller et al., 2022? This should be discussed in the revised manuscript.

The reviewer has raised an important point. Indeed, the EuMP cycle has the same theoretical production yields as the natural RuMP. As there is no kinetics data on the isomerases (the assays are limited by substrate unavailability and assay method), it is hard to predict the maximal growth rate via this pathway and compare it to the RuMP. Still, the natural methylotrophs *B. methanolicus* MGA3 grows on methanol with growth rates of 0.14 h⁻¹ at 37°C and 0.46 h⁻¹ at 50°C (Ref DOI: 10.1128/mSystems.00745-20 & 10.1002/pmic.201300515), the synthetic methylotrophic *E. coli* currently has only growth rate of 0.087 h⁻¹ (doubling time of 8 h) after years extensive studies by different laboratories. The presented EuMP cycle would also require similar more extensive research to establish. Nevertheless, we believe it holds the promise as an efficient C1-platform in future bioindustry once significant rates and yields are established.

We revised the discussion:

“In this study, we present a new-to-nature formaldehyde assimilation pathway, the EuMP cycle, following a mix-and-match engineering strategy similar to the homoserine cycle. Our theoretical calculations highlight that the EuMP cycle is equally efficient as the RuMP cycle, outperforming other natural C1-assimilation pathways. Although the EuMP cycle requires more enzymes than the RuMP cycle, it is hard to compare the enzyme costs of both pathways because of the difficulties of obtaining full kinetics of the core enzymes. Still the EuMP cycle serves as a complementation to the synthetic homoserine cycle, which has relatively low theoretical yields in biomass and sugar phosphates. Moreover, the EuMP cycle holds the potential to be further extended by phosphoketolase reaction to further improve its efficiency, as phosphoketolase produces C2 moieties from sugar phosphates without oxidation (releasing CO₂).”

On the other hand, as our response to reviewer 3, our study may also pinpoint a hidden natural methylotrophic pathway. This cycle may exist in some microbes that have the erythritol catabolism enzymes (the pentose phosphate pathway enzymes are wide spread). We did not stress this point in the manuscript to avoid misleading because we do not have direct data, but preliminary results that do not support this in our tested organism (we still cannot rule out the latent existence of EuMP in the strain). We added to the discussion:

“It was suggested that EPS in *P. pentosaceum* and *Sinorhizobium meliloti* functions in the Eu1P cleavage direction for erythritol catabolism. Further study on the co-occurrence of EPS and these isomerases (as well as the other tetritol catabolism enzymes) is particularly interesting, because it could not only elucidate their physiological functions from evolutionary and ecological aspects, but may also find that the EuMP cycle is already a naturally existing C1-assimilation pathway, similar to the reductive glycine pathway.”

Point 2: In the revised manuscript, the authors should better explain their choice on the origin of the genes encoding for the enzymes of the M2 module as it has been done for the EPS encoding genes. Indeed, authors choose to express the genes from *M. smegmatis* but those enzymes are also found in others bacteria (e.g. *B. abortus*) as mentioned in BioCyc. Did the authors tried other genes encoding enzymes?

As the enzymes in M2 are responsible for their native/primary activities, for the pathway demonstration we did not try other homologs. Indeed, testing other homologs to find better enzyme would be beneficial, thus we added in the discussion:

“Similarly, better enzymes for the isomerases, i.e., EryC, LerI and DerI, can also be identified and substituted. Microorganisms catabolizing tetritols offers a broad mining source.”

Point 3: Figure 2 shows that the plasmid pKIID rescue growth of the Δ tktAB strain on 10 mM L-threitol and erythritol, but the growth starts after a 40 to 50 hours of lag phase. How can it be explained? What happens during this long lag phase?

We thank the reviewer for the comment. To explain this growth phenotype, we added Supplementary Figure S3. Our growth experiment result indicated that while both 10 mM L-threitol and erythritol were not toxic, their catabolism intermediates caused growth lag. It's known that erythrulose can react with amino acids in proteins, a similar chemical property as dihydroxyacetone. This is likely the cause of the lag phase. We added in the text:

“We note that the lag phase seems to result from tetritol catabolism intermediates, it appeared only when both the pKIID plasmid and L-threitol or erythritol presented (Supplementary Fig. S3).”

We note that we changed the colors and line styles in Fig. 2b to follow the same scheme as in the Supplementary Fig. S3. We also removed a previously duplicated negative control in Fig. 2b.

Point 4: L185: Based on the figure 3, authors claim that no growth was observed without sarcosine in the strains overexpressing pFEIIO or PFEIIO plasmid but in these conditions (red line figure 3c&d) a small growth is observed. How can it be explained?

We thank the reviewer for the comment. This is the same comment as the first major comment from reviewer #1. Indeed, we observed the phenomenon in multiple replicated experiments. However, only if the full pathway was overexpressed. Hence, we attribute it to the presence of internal source formaldehyde. Our hypothesis can be supported by evidences: (1) formaldehyde is non-depletable from cellular metabolism, many biological processes produce this compound (Ref: 9 & 11); (2) such growth was still pathway enzymes dependent; (3) relative too high fractions of unlabeled Phe and Tyr (Fig. 3h); (4) both FucA and RhaD supported growth under a low sarcosine condition (0.2 mM, orange lines) and at the same time such growth was not observed in the higher selection Δ FBP/GlpX strain. We revised the text:

“In these strains, growth rates and maximal ODs directly correlated with increasing sarcosine concentrations (Fig. 3 c and d) and growth could not be restored without EPS (i.e., *fucA* or *rhaD*) overexpression (Fig. 3e). ... We attributed the small growth under sarcosine absent condition (red lines in Fig. 3 c and d) to a non-depletable internal source of formaldehyde, because such growth was FucA/RhaD and EryC dependent (Fig. 3 e, f and Supplementary Fig. S5).”

“...the relatively high unlabeled fractions, especially for FucA, were likely from internal source formaldehyde originated from unlabeled glycerol.”

Point 5: In figures 3 h and 4 d, there is always few amounts of labelling that is found in Alanine although no labelled should be found as is it claimed in L212 and 259. The same comments can be done for the labelling pattern of tyrosine and phenylalanine which are supposed to be single-labeled as claimed in L 208 and 259 but where few double-labeled is observed. How can it be explained?

The labeling in alanine as well as double-labeled tyrosine and phenylalanine can be explained by natural ^{13}C isotope occurrence. Because we used normal glycerol, not ^{13}C - depleted $^{12}\text{C}_3$ -glycerol, the ambient ^{13}C from glycerol would add up a small fraction of higher labeling. Such ambient ^{13}C fractions are also reflected in the Supplementary Fig. S6, where unlabeled sarcosine was used. In order to clarify this to the readers, we thus added in the text:

“The small fractions of double labeling were results of natural abundance of ^{13}C (Supplementary Fig. S6a) ...”

Point 6: L341: authors said that: “The in vivo establishment of the full EuMP cycle relying on formaldehyde solely will require more metabolic engineering and long-term ALE”. However, short ALE experiments performed in this work demonstrated that *E. coli* was able to partially bypass the EuMP cycle after 5 passages of evolution. Isn't the cycle likely to be completely hijacked with longer ALE experiments? This point should be discussed in the revised manuscript.

We thank the reviewer for the comment. For the long-term ALE experiments towards the establishment of the full cycle using formaldehyde, we believe that there are selection options that can provide a good start for such an experiment, which are unlikely to get hijacked. Metabolism, apparently provides several options, e.g. for converting sugar phosphates to lower glycolysis, but these reactions are usually one-directional. Hence hijacking would be also unlikely to happen in the proposed upper metabolism selection (glycolysis is blocked). Because similar selections used (as by ref 64) did not observe any breakage from lower metabolism in their long term ALE.

We added to the discussion:

“Although we observed partial bypass in the FUI ALE, similar occurrences restoring growth are unlikely in such a full cycle selection condition which has a selection pressure similar to the abovementioned examples.”

Minor:

L100: “and NAD(P)H” can be removed as the mentioned cycles do not use it.

We thank the reviewer for the correction.

Fig 1. The EryC reaction should be added in the schematic representation of the EuMP cycle.

We thank the reviewer's comment. We modified the figure and revised the text:

“The key reaction, catalyzed by erythrose 1-phosphate synthase (EPS, or erythrose 1-phosphate formaldehyde-lyase), condenses formaldehyde with DHAP yielding erythrose 1-phosphate (Eu1P). Depending

on the enzyme stereoselectivity, the product could be L-Eu1P or D-Eu1P, the latter one can be converted by D-erythrulose 1-phosphate 3-epimerase (EryC) into L-Eu1P.”

Figure 3 g: unlabeled PEP can also derive from the GAP originating from TAL reaction. This should be added.

The reviewer is indeed correct. All unlabeled carbons are originally derived from glycerol/DHAP, ignoring glycine carbons from sarcosine oxidation. We reasoned that formaldehyde derived carbon on E4P is transferred to S7P in TAL reaction (Refer to Fig. 5a) and there is no S7P sink resulting a net 0 flux of TAL (calculated from flux balance analysis and added Supplementary Fig. S7a), to avoid a complex but seemingly not helpful scheme, we added to the text:

“We note that TAL transfers carbons between F6P and GAP, resulting in unlabeled GAP and subsequently PEP. Yet the flux of TAL is net 0 (calculated by flux balance analysis, Supplementary Fig. S7a).”

Reviewer #3 (Remarks to the Author):

He et al. devised a new-to-nature formaldehyde fixation pathway and embedded it into *E. coli* for efficient C1 assimilation. This pathway synthetic formaldehyde assimilation pathway was named the erythrulose monophosphate (EuMP) cycle and it relies on the side activity of dihydroxyacetone phosphate (DHAP) dependent aldolase. This erythrulose 1-phosphate synthase reaction condenses formaldehyde with DHAP to yield L-erythrulose 1-phosphate. By two isomerase reactions, the latter can be converted to D-erythrose 4-phosphate (E4P), an intermediate of the pentose phosphate shunt. Conversion of E4P by reactions of the pentose phosphate shunt, triosephosphate isomerase, phosphofruktokinase and fructose bisphosphate aldolase regenerates DHAP. In addition, three further variants involving sedoheptulose 1,7-bisphosphate adolase and/or the Entner-Doudoroff variant of glycolysis. However, the chosen transaldolase/fructose bisphosphate aldolase is the most energy efficient variants.

Embedding into *E. coli* metabolism in vivo, was followed by gradual implementation of 4 modules. First, E4P was isolated by deletion of transketolase genes and a mutant expressing genes of tetrytol catabolism provided E4P from either erythritol or threitol. This demonstrated that isomerase LerI and DerI provided sufficient E4P for growth of *E. coli* (module 2). Next, erythrulose 1-phosphate dependent formaldehyde fixation was established. Side activities of a number of aldolase regarding the aldehyde acceptor substrate (but not the donor substrate) were sought in literature for the aldolases depending on DHAP as donor. Formaldehyde as acceptor substrate was generated from sarcosine by sarcosine oxidase (in situ, adjustable in a strain lacking formaldehyde oxidation to formate). Of several candidates, L-fuculose 1-phosphate aldolase (FucA) from *E. coli* performed best to assimilate formaldehyde (module 1). Tracer experiments support this notion. Not surprinsingly, the native pentose phsosphate shunt enzymes did not operate to allow for efficient integration of the modules 1 and 2 and made adaptive laboratory evolution (ALE) necessary. First, a double deletion lacking both fructose bisphosphatases was generated which cannot grow with glycerol as sole carbon source as E4P, ribose 5-phosphate and glucose 6-phosphate cannot be generated. However, when modules 1 and 2 operate, these intermediates can be generated from sarcosine and glycerol. Two independent ALE mutants were selected and

grew well with glycerol and sarcosine. Tracer experiments support this notion. Genome sequencing of the ALE mutants showed that these shared only one mutation: a SNP in the promoter region of *psuK* that codes for pseudouridine kinase, an enzyme required for degradation of pseudouridine, a nucleoside found, e.g., in tRNAs. It remains unclear if and how the promoter activity of *psuK* was affected or if this SNP is at all relevant for the selected growth phenotypes of ALE mutants FUI and FUII.

In FUII, mutated proteins TpiA and LerI provided a likely explanation for the observed phenotype. In FUI, the requirement for fructose 6-phosphate was lowered. Would introduction of the SNPs in the TpiA and LerI genes in FUI improve its growth with glycerol and sarcosine?

We thank the reviewer for the suggestion. We tried to reconstruct the TpiA G94D in the FUI strain with different methodologies, but unfortunately, all our trials failed and we did not manage to construct the strain. Although introducing the mutation into the parental Δ FBP/GlpX strain worked. Since it is beyond pathway demonstration, the main message of the current manuscript, we ask the reviewer's kind understanding for not answering by providing experimental results.

We did transform the plasmid with a mutated *lerI*, pFEIIOe, into the naïve strain and FUI. We added the results along with other characterization results in Supplementary Figure S9 and in the text:

“Indeed, the importance of both LerI and TpiA mutations was confirmed. Plasmid curing then re-transforming the plasmid pFEIIO (with naïve LerI, Supplementary Table S1) abolished growth of FUII strain (Supplementary Fig. S9a). FUII strain reverted the *tpiA* mutation to wild-type (FUII-*tpiA*_wt, Supplementary Table S1) did not grow in the presence of the plasmid with a mutated LerI (pFEIIOe, Supplementary Table S1 and Supplementary Fig. 9b). Although introducing a mutation of either LerI or TpiA alone was not sufficient to enable the growth of the parental Δ FBP/GlpX strain, together, they allowed growth of the strain (Supplementary Fig. S9). These results indicate synergy between mutations in LerI and TpiA. Finally, considering the regulation role of *PsuK* (as mentioned above), the extended lag phase observed in the reverse-engineered Δ FBP/GlpX-*tpiA*_m strain with pFEIIOe, compared to the FUII strain, may suggest an additional layer of synergy with *psuK* promoter mutation (Supplementary Fig. S9b and Fig. 4b).”

The authors do not discuss module 4 much: why?

There are three reasons: 1) the module 4 is part of glycolysis, thus should be active; 2) unlike the module 3, which requires reconfiguration of the complex pentose phosphate pathway, module 4 is linear and operates in the same direction as when feeding with glucose; 3) we use a stepwise engineering strategy, module 4 will be needed in the next stage of establishing the whole cycle. We added to the discussion:

“... Module M4 should be active itself as it is part of glycolysis and operates in the same direction as when feeding with glucose, integrating it with the other three modules is still not an easy task ...”

The enzymes with erythrose 1-phosphate synthase side activity were important to establish this new-to-nature cycle. Is it actually new-to-nature? May be the donor microorganisms operate this or a related cycle for

formaldehyde assimilation. Do these enzymes occur in combination with enzymes of tetritol catabolism? Sarcosine oxidase? Can we learn from evolutionary considerations and absence/occurrence of these side activities?

The reviewer has raised a very interesting point here. The EuMP cycle might exist in nature like the reductive glycine pathway that was found in nature after synthetically established (Ref: Sánchez-Andrea, et al., 2020. The reductive glycine pathway allows autotrophic growth of *Desulfovibrio desulfuricans*. Nature Communications 11, 5090.). However, there is no report indicating such existence of the EuMP pathway from our knowledge, therefore in our eyes it is a new-to-nature pathway. Indeed, EPS was found to co-exist with erythritol catabolism enzymes within *Propionibacterium pentosaceum* (Wawszkiewicz & Barker, 1968. Erythritol Metabolism by *Propionibacterium pentosaceum*. Journal of Biological Chemistry 243, 1948–1956.) and *Sinorhizobium meliloti* (Geddes et al., 2010. A locus necessary for the transport and catabolism of erythritol in *Sinorhizobium meliloti*. Microbiology 156, 2970–2981.), but both studies suggested the physiological function of EPS of cleaving Eu1P to formaldehyde and DHAP, to be used in metabolism.

We added in the discussion section:

“It was suggested that EPS in *P. pentosaceum* and *Sinorhizobium meliloti* functions in the Eu1P cleavage direction for erythritol catabolism. Further study on the co-occurrence of EPS and these isomerases (as well as the other tetritol catabolism enzymes) is particularly interesting, because it could not only elucidate their physiological functions from evolutionary and ecological aspects, but may also find that the EuMP cycle is already a naturally existing C1-assimilation pathway, similar to the reductive glycine pathway.”

In prior experiments, formaldehyde was shown to react chemically with DNA and the appropriate compensatory mutations were selected. Here, these genes were not mutated. Is this due to the sarcosine strategy of providing formaldehyde?

As the reviewer pointed out, formaldehyde can damage DNA/protein via crosslinking. The significance of such effect would rely on the concentration/dose of formaldehyde. Here we wouldn't expect a high level formaldehyde as we provided only 1 mM sarcosine, hence no high formaldehyde accumulations are expected, an advantage when using this *in situ* formaldehyde generation strategy. On the other hand, FucA supported growth of the Δ frmTKT selection even under 0.2 mM sarcosine, indicating it's a good sink of formaldehyde. Taking together, formaldehyde concentrations might not be high enough to induce gene mutations.

I. 104: FBA and FBA: enzyme or flux balance analysis?

We apologize for the confusion. Here “FBA” represents flux balance analysis. We now revised “flux balance analysis” by not using abbreviations throughout the text. “FBA” represents only the enzyme, fructose-bisphosphate aldolase, in the text now.

Similarly, we revised abbreviation of “transaldolase” all to “TAL” where it was “TA”.

Fig. 2: the abbreviation d is often used for desoxy: why not rename dEu4P to DEu4P or D-Eu4P (similar lEu1P to L-Eu1P)?

Following the reviewer's comment, we changed in the text and figures the abbreviations of "dEu1P", "lEu1P", "dEu4P", "dEu", and "lEu" to "D-Eu1P", "L-Eu1P", "D-Eu4P", "D-Eu", and "L-Eu", respectively.

I.134 what is E4P_suppl.?

"E4P_suppl." represents E4P (erythrose 4-phosphate) supplements. As described in the main text, it contains E4P derived shikimate, aromatic amino acids and vitamins, as well as pyridoxine. The full composition is described in the Method section. We added the abbreviations to the corresponding position.

Reviewers' Comments:

Reviewer #1:

Remarks to the Author:

This study deployed the efficient formaldehyde assimilation cycle based on promiscuous dihydroxyacetone phosphate-dependent aldolase. The energy efficiency of this novel pathway was demonstrated via in silico modeling. Furthermore, the in vivo feasibility was demonstrated with various supporting data. This work contributed a significant concept for the further development of synthetic methylotrophs. The revised version covered the comments of other reviewers. However, some comments need to be covered for this manuscript which are shown below.

Major

1. As mentioned in the manuscript, this work deployed the sarcosine as the source of formaldehyde instead of C1 carbon sources. Thus, the abstract should be revised to clarify this.
2. The scenario in this study is to deploy the most efficient pathway variant TAL/FBA with the focus of engineering places on the two first modules of the total four modules. Thus, is there any possibility that the flux of intermediates produced from module 2 goes into the less efficient variant of the two latter modules (SBP/FBA, TAL/ED, or SBP/ED)? If so, are there any suggestions to unravel this situation?
3. Line 179-180. Even though the EryC reaction is fully reversible, is there any possibility that the burden of genetic engineering and heterologous expression of this gene could cause some impacts on the pathway efficiency when combined with FbaA and YihT? Could this be part of the reason why these strains failed to grow?
4. Line 237. The Δ FBP/GlpX strain eliminated the carbon flux to F6P from DHAP which resulted in more dependence on another carbon source (xylose). Even though the ALE experiment helped to reduce the dependence on xylose, the mutation characterization showed that the strain adapted to the limiting conditions instead of enhancing the assimilation efficiency of the introduced modules. Thus, does this mean that the operation of M1, M2, and M3 is not efficient enough for the mutated E. coli to maintain its metabolism?
5. Line 390-392. The revised statement "decoupling the challenge of formaldehyde production from C1" showed some confusion in the same sentence with the following phrase. Please consider revising the whole sentence for conciseness.

Minor

1. Line 75: Revise "Overall" to "Overall,"
2. Line 323. "in FUII (Supplementary Fig. S1 a and b) in FUII". Typo mistake.
3. Line 414. "Gene-knocks" is knock-in or knock-out?
4. Line 452. E4P_suppl. should be notified at the first mention in the manuscript instead of at the latter section "Materials and methods".

Reviewer #2:

Remarks to the Author:

All my revisions have been considered in this revised version and answers are convincing. These additions clearly enhance the impact of the work. The manuscript is publishable as is.

Reviewer #3:

Remarks to the Author:

The authors improved their manuscript and added new data. I am satisfied with their revision.

Reviewer #1 (Remarks to the Author):

This study deployed the efficient formaldehyde assimilation cycle based on promiscuous dihydroxyacetone phosphate-dependent aldolase. The energy efficiency of this novel pathway was demonstrated via *in silico* modeling. Furthermore, the *in vivo* feasibility was demonstrated with various supporting data. This work contributed a significant concept for the further development of synthetic methylotrophs. The revised version covered the comments of other reviewers. However, some comments need to be covered for this manuscript which are shown below.

We thank the reviewer's support and valuable scientific inputs.

Major

1. As mentioned in the manuscript, this work deployed the sarcosine as the source of formaldehyde instead of C1 carbon sources. Thus, the abstract should be revised to clarify this.

We thank the reviewer for the comment. We thus have revised the abstract accordingly.

We further adjusted the entire abstract to adhere the required word limit (now 147 words).

2. The scenario in this study is to deploy the most efficient pathway variant TAL/FBA with the focus of engineering places on the two first modules of the total four modules. Thus, is there any possibility that the flux of intermediates produced from module 2 goes into the less efficient variant of the two latter modules (SBP/FBA, TAL/ED, or SBP/ED)? If so, are there any suggestions to unravel this situation?

Yes, since *E. coli* natively contains enzymes of the other variants, it is possible that the flux goes into the less efficient variants. Our results indeed indicated the operation of the SBP variant, as discussed in line 306 - 309.

The ED variant can be blocked by gene knock-out, for example *zwf*. The SBP variant is difficult to block because, the key reaction catalyzed by FbaA (DOI: 10.1038/msb.2009.65), whose primary activity is required in the FBA route, and likely various phosphatases (a similar situation to fructose-1,6-bisphosphatase), noting that we observed SBP route activity although we have knocked out *glpX*, a reported SBPase (DOI: 10.1038/s41467-018-04795-4).

However, although our demonstration focused on the most efficient TAL/FBA variant, the presence of the other variants can provide metabolic flexibility to adapt different C1 feedstock and production scenarios providing robustness. We herein added in the discussion:

“EuMP can operate in four different modes, offering metabolic flexibility and potentially enhancing adaptability to various requirements.”

We added also our speculation in the result section about the SBPase:

“Although the strain does not contain GlpX, an identified sedoheptulose 1,7-Bisphosphatase, other phosphatases may possess such activity, similar to the aforementioned SBP.”

3. Line 179-180. Even though the EryC reaction is fully reversible, is there any possibility that the burden of genetic engineering and heterologous expression of this gene could cause some impacts on the pathway efficiency when combined with FbaA and YihT? Could this be part of the reason why these strains failed to grow?

We thank the reviewer for the comment. Indeed, we could not completely rule out the possibility that expression burden was part of the reason for the no growth phenotype of FbaA and YihT. However, we believe it is not the major reason, reasoning that (1) our plasmid system has relatively low burden to the cells: p15A has ~10 copies/cell (DOI: 10.1038/s41467-021-21734-y) and the promoter was derived from *E. coli* *pgi* promoter for physiological expression; (2) we got similar growth for RhaD between with and without EryC expression, as shown in Figure 3 d and f, respectively, indicating no growth inhibitory effects between the strains.

4. Line 237. The Δ FBP/GlpX strain eliminated the carbon flux to F6P from DHAP which resulted in more dependence on another carbon source (xylose). Even though the ALE experiment helped to reduce the dependence on xylose, the mutation characterization showed that the strain adapted to the limiting conditions instead of enhancing the assimilation efficiency of the introduced modules. Thus, does this mean that the operation of M1, M2, and M3 is not efficient enough for the mutated *E. coli* to maintain its metabolism?

We thank the reviewer for the comment. While evolved strain FUI lowered the selection pressure, mutations in TpiA and LerI within FUII enhanced efficiency of the EuMP modules and fulfilled the requirements for growth of the selection strain. Our reverse-engineered strain grew under sarcosine (and glycerol, without xylose) condition (Supplementary Figure S9b, solid green line).

5. Line 390-392. The revised statement “decoupling the challenge of formaldehyde production from C1” showed some confusion in the same sentence with the following phrase. Please consider revising the whole sentence for conciseness.

We thank the reviewer for the correction. We rephrased the sentence:

“These strategies could enable the growth of *E. coli* via the full EuMP cycle first on sarcosine-derived formaldehyde, decoupling the challenge of formaldehyde production from methanol, formate, or other C1 feedstocks, paving the way for the future development of growth solely on C1 feedstocks.”

Minor

1. Line 75: Revise “Overall” to “Overall,”

Corrected.

2. Line 323. “in FUII (Supplementary Fig. S1 a and b) in FUII”. Typo mistake.

Corrected.

3. Line 414. “Gene-knocks” is knock-in or knock-out?

We thank the reviewer for pointing out. We corrected in the text to “gene knock-outs”.

4. Line 452. E4P_suppl. should be notified at the first mention in the manuscript instead of at the latter section “Materials and methods”.

We thank the reviewer for pointing out. We notified at the first mention and rephrased the sentence to have better flow.

“To grow on minimal medium, this strain requires E4P supplements (E4P_suppl.: 1 mM shikimic acid, 1 μ M pyridoxine, 250 μ M tyrosine, 500 μ M phenylalanine, 200 μ M tryptophan, 6 μ M 4-aminobenzoic acid, 6 μ M 4-hydroxybenzoic acid and 50 μ M 2,3-dihydroxybenzoic acid), because E4P is a precursor of several essential cellular building blocks and cofactors, such as shikimate derived aromatic amino acids and vitamins as well as pyridoxal phosphate.”

Reviewer #2 (Remarks to the Author):

All my revisions have been considered in this revised version and answers are convincing. These additions clearly enhance the impact of the work. The manuscript is publishable as is.

We thank the reviewer’s support and valuable scientific inputs.

Reviewer #3 (Remarks to the Author):

The authors improved their manuscript and added new data. I am satisfied with their revision.

We thank the reviewer’s support and valuable scientific inputs.

Reviewers' Comments:

Reviewer #1:

Remarks to the Author:

All the comments to the previous manuscript is properly revised in the revised manuscript. I recommend this manuscript to be published without further revision.

Reviewer #1 (Remarks to the Author):

All the comments to the previous manuscript is properly revised in the revised manuscript. I recommend this manuscript to be published without further revision.

We thank the reviewer for support and valuable scientific inputs.